# Extracellular electron transfer-dependent anaerobic oxidation of ammonium by anammox bacteria

Dario R. Shaw[1], Muhammad Ali[1], Krishna P. Katuri[1], Jeffrey A. Gralnick [2], Joachim Reimann[3], Rob Mesman [3], Laura van Niftrik [3], Mike S. M. Jetten[3] & Pascal E. Saikaly [1✉]

Anaerobic ammonium oxidation (anammox) bacteria contribute significantly to the global nitrogen cycle and play a major role in sustainable wastewater treatment. Anammox bacteria convert ammonium ($NH_4^+$) to dinitrogen gas ($N_2$) using intracellular electron acceptors such as nitrite ($NO_2^-$) or nitric oxide (NO). However, it is still unknown whether anammox bacteria have extracellular electron transfer (EET) capability with transfer of electrons to insoluble extracellular electron acceptors. Here we show that freshwater and marine anammox bacteria couple the oxidation of $NH_4^+$ with transfer of electrons to insoluble extracellular electron acceptors such as graphene oxide or electrodes in microbial electrolysis cells. $^{15}N$-labeling experiments revealed that $NH_4^+$ was oxidized to $N_2$ via hydroxylamine ($NH_2OH$) as intermediate, and comparative transcriptomics analysis revealed an alternative pathway for $NH_4^+$ oxidation with electrode as electron acceptor. Complete $NH_4^+$ oxidation to $N_2$ without accumulation of $NO_2^-$ and $NO_3^-$ was achieved in EET-dependent anammox. These findings are promising in the context of implementing EET-dependent anammox process for energy-efficient treatment of nitrogen.

[1] Water Desalination and Reuse Center (WDRC), Biological and Environmental Science & Engineering (BESE) Division, King Abdullah University of Science and Technology (KAUST), Thuwal 23955-6900, Saudi Arabia. [2] BioTechnology Institute and Department of Plant and Microbial Biology, University of Minnesota, Twin Cities, St. Paul, MN 55108, USA. [3] Department of Microbiology, Institute for Water and Wetland Research (IWWR), Faculty of Science, Radboud University, Heyendaalseweg 135, 6525 AJ Nijmegen, The Netherlands. ✉email: pascal.saikaly@kaust.edu.sa

Anaerobic ammonium oxidation (anammox) by anammox bacteria contributes up to 50% of $N_2$ emitted into Earth's atmosphere from the oceans[1,2]. Also, anammox bacteria have been extensively investigated for energy-efficient removal of $NH_4^+$ from wastewater[3]. Initially, anammox bacteria were assumed to be restricted to $NH_4^+$ as electron donor and $NO_2^-$ or NO as electron acceptor[4,5]. More than a decade ago, preliminary experiments suggested that *Kuenenia stuttgartiensis* and *Scalindua* could couple the oxidation of formate to the reduction of insoluble extracellular electron acceptors such as Fe(III) or Mn (IV) oxides[6,7]. However, extracellular electron transfer (EET) activity and the molecular mechanism of this coupling reaction have remained unexplored to date. Further, these tests with *K. stuttgartiensis* and *Scalindua* could not discriminate between Fe (III) oxide reduction for nutritional acquisition (i.e., via siderophores) vs. respiration through EET[8]. Therefore, these preliminary experiments are not conclusive to determine if anammox bacteria have EET capability or not.

Although preliminary work showed that *K. stuttgartiensis* could not reduce Mn(IV) or Fe(III) with $NH_4^+$ as electron donor[6], the possibility of anammox bacteria to oxidize $NH_4^+$ coupled to EET to other type of insoluble extracellular electron acceptors cannot be ruled out. In fact, EET (and set of genes involved with EET) is not uniformly applied to all insoluble extracellular electron acceptors; some electroactive bacteria are not able to transfer electrons to carbon-based insoluble extracellular electron acceptors such as electrodes in bioelectrochemical systems but could reduce metal oxides and vice versa[9]. It has been known for more than two decades that carbon-based high-molecular-weight organic materials, which are ubiquitous in terrestrial and aquatic environments and that are not involved in microbial metabolism (i.e., humic substances) can be used as an external electron acceptor for the anaerobic oxidation of compounds[10]. Also, it has been reported that anaerobic $NH_4^+$ oxidation linked to the microbial reduction of natural organic matter fuels nitrogen loss in marine sediments[11]. A literature survey of more than 100 EET-capable species indicated that there are many ecological niches for microorganisms able to perform EET[12]. This resonates with a recent finding where *Listeria monocytogenes*, a host-associated pathogen and fermentative gram-positive bacterium, was able to respire through a flavin-based EET process and behaved as an electrochemically active microorganism (i.e., able to transfer electrons from oxidized fuel (substrate) to a working electrode via EET process)[13]. Further, it was reported that anammox bacteria seem to have homologs of *Geobacter* and *Shewanella* multi-heme cytochromes that are responsible for EET[14]. These observations stimulated us to investigate whether anammox bacteria can couple $NH_4^+$ oxidation with EET to carbon-based insoluble extracellular electron acceptor and can behave as electrochemically active bacteria.

Here we report that in the absence of $NO_2^-$, phylogenetically distant anammox bacteria couple the anaerobic oxidation of $NH_4^+$ with transfer of electrons to carbon-based insoluble extracellular electron acceptors such as graphene oxide (GO) or electrodes poised at a certain potential in microbial electrolysis cells (MECs). Our results also revealed that anammox bacteria oxidized $NH_4^+$ to $N_2$ with $NH_2OH$ as intermediate of the process. Interestingly, the electrons released from the $NH_4^+$ oxidation were transferred to the extracellular electron acceptor via a pathway that is analog to the ones present in metal-reducing organisms such as *Geobacter* spp. and *Shewanella* spp. Taken together, our results revealed the potential of anammox bacteria to use solid-state electron acceptors as the terminal electron sink and demonstrated that there is no need for $NO_2^-$, $NO_3^-$ or partial nitritation for anaerobic $NH_4^+$ oxidation.

## Results and discussion

**Ammonium oxidation coupled with EET**. To evaluate if anammox bacteria possess EET capability, we first tested whether enriched cultures of three phylogenetically and physiologically distant anammox species can couple the oxidation of $NH_4^+$ with the reduction of an insoluble extracellular electron acceptor. Cultures of *Ca.* Brocadia (predominantly adapted to freshwater environments) and *Ca.* Scalindua (predominantly adapted to marine water environments) were enriched and grown as planktonic cells in membrane bioreactors (Supplementary Fig. 1a)[15]. Fluorescence in situ hybridization (FISH) showed that the anammox bacteria constituted >95% of the bioreactor's community (Supplementary Fig. 1b–g). Also, a previously enriched *K. stuttgartiensis* (predominantly adapted to freshwater environments) culture was used[4]. The anammox cells were incubated anoxically for 216 h in the presence of $^{15}NH_4^+$ (4 mM) and GO as a proxy for insoluble extracellular electron acceptor. No $NO_2^-$ or $NO_3^-$ was added to the incubations. GO particles are bigger than bacterial cells and cannot be internalized, and thus GO can only be reduced by EET[16]. Indeed, GO was reduced by anammox bacteria as shown by the formation of suspended reduced GO (rGO), which is black in color and insoluble (Fig. 1a)[16]. In contrast, abiotic controls did not form insoluble black precipitates. Reduction of GO to rGO by anammox bacteria was further confirmed by Raman spectroscopy, where the formation of the characteristic 2D and D + D′ peaks of rGO[17] were detected in the vials with anammox cells (Fig. 1b), whereas no peaks were detected in the abiotic control. Further, isotope analysis of the produced $N_2$ gas showed that anammox cells were capable of $^{30}N_2$ formation (Fig. 1c). In contrast, $^{29}N_2$ production was not significant in any of the tested anammox species or controls, suggesting that unlabeled $NO_2^-$ or $NO_3^-$ were not involved. The production of $^{30}N_2$ indicated that the anammox cultures use a different mechanism for $NH_4^+$ oxidation in the presence of an insoluble extracellular electron acceptor (further explained below). Gas production was not observed in the abiotic control (Fig. 1c). To determine if anammox bacteria were still dominant after incubation with GO, we extracted and sequenced total DNA from the *Brocadia* and *Scalindua* vials at the end of the experiment. Differential coverage showed that the metagenomes were dominated by anammox bacteria (Supplementary Fig. 2a, c). Taken together, these results support that anammox bacteria have EET capability.

**Electroactivity of anammox bacteria**. Electrochemical techniques provide a powerful tool to evaluate EET, where electrodes substitute for the insoluble minerals as the terminal electron acceptor[13]. Compared with metal oxides, the use of electrodes as the terminal electron acceptor allow us to quantify the number of externalized electrons per mol of $NH_4^+$ oxidized. Also, since the electrode is only used for bacterial respiration, then we can better assess EET activity compared with metal oxides, where we cannot differentiate between metal oxide reduction for nutritional acquisition from respiration through EET activity. Therefore, we tested if anammox bacteria interact with electrodes via EET and use them as the sole electron acceptor in MEC. One single-chamber MEC operated at eight different set potentials (from −0.1 to 0.6 V vs. standard hydrogen electrode (SHE)) using multiple working electrodes (Supplementary Fig. 1h) was initially operated under abiotic conditions with the addition of $NH_4^+$ only. No current and $NH_4^+$ removal were observed in any of the abiotic controls. Subsequently, the *Ca.* Brocadia culture was inoculated into the MEC and operated under optimal conditions for anammox (i.e., addition of $NH_4^+$ and $NO_2^-$). Under this scenario, $NH_4^+$ and $NO_2^-$ were completely removed from the

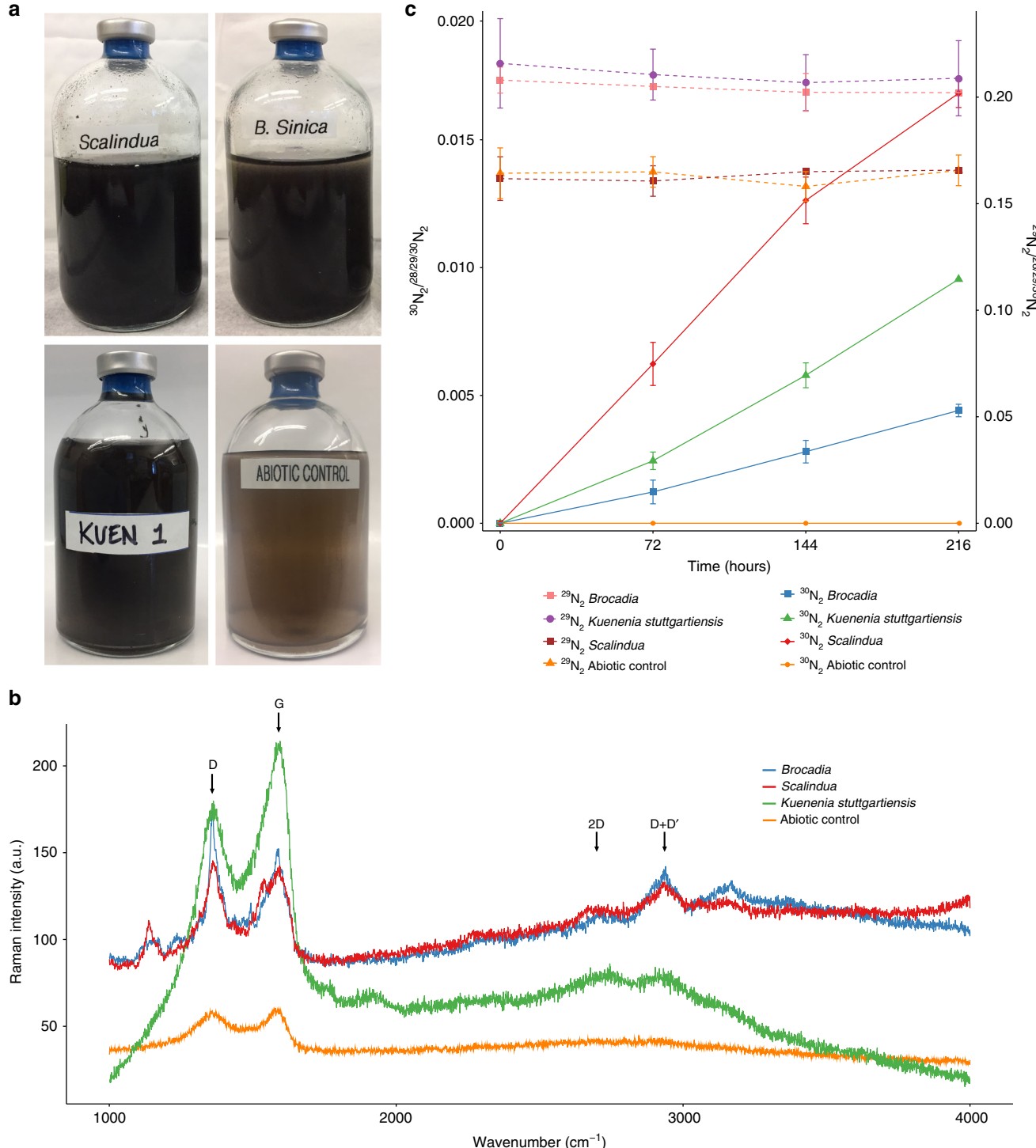

**Fig. 1 NH$_4^+$ oxidation coupled with extracellular electron transfer. a** Photographs of serum vials after 216 h of incubation with different species of anammox bacteria, $^{15}$NH$_4^+$, and graphene oxide (GO). The presence of black precipitates indicates the formation of reduced GO (rGO). No obvious change in color was observed in the abiotic control vials after the same period of incubation with $^{15}$NH$_4^+$ and GO. **b** Raman spectra of the vials after 216 h of incubation. Peaks in bands of 2D and D + D′ located at ~2700 and ~2900 cm$^{-1}$, respectively, indicate the formation of rGO. **c** $^{30}$N$_2$ production by different anammox bacteria from $^{15}$NH$_4^+$ and GO as the sole electron acceptor. Anammox cells were incubated with 4 mM $^{15}$NH$_4^+$ and GO to a final concentration of 200 mg L$^{-1}$. There was no $^{29}$N$_2$ formation throughout the experiment. NO and N$_2$O were not detected throughout the experiment. Results from triplicate serum vial experiments are represented as mean ± SD.

medium without any current generation (Fig. 2a). Stoichiometric ratios of consumed NO$_2^-$ to consumed NH$_4^+$ ($\Delta$NO$_2^-$/$\Delta$NH$_4^+$) and produced NO$_3^-$ to consumed NH$_4^+$ ($\Delta$NO$_3^-$/$\Delta$NH$_4^+$) were in the range of 1.0–1.3 and 0.12–0.18, respectively, which are close to the theoretical ratios of the anammox reaction[18]. These

ratios indicated that anammox bacteria were responsible for NH$_4^+$ removal in the MEC. Subsequently, NO$_2^-$ was gradually decreased to 0 mM leaving the electrodes as the sole electron acceptor. When the exogenous electron acceptor (i.e., NO$_2^-$) was completely removed from the feed, anammox cells began to form

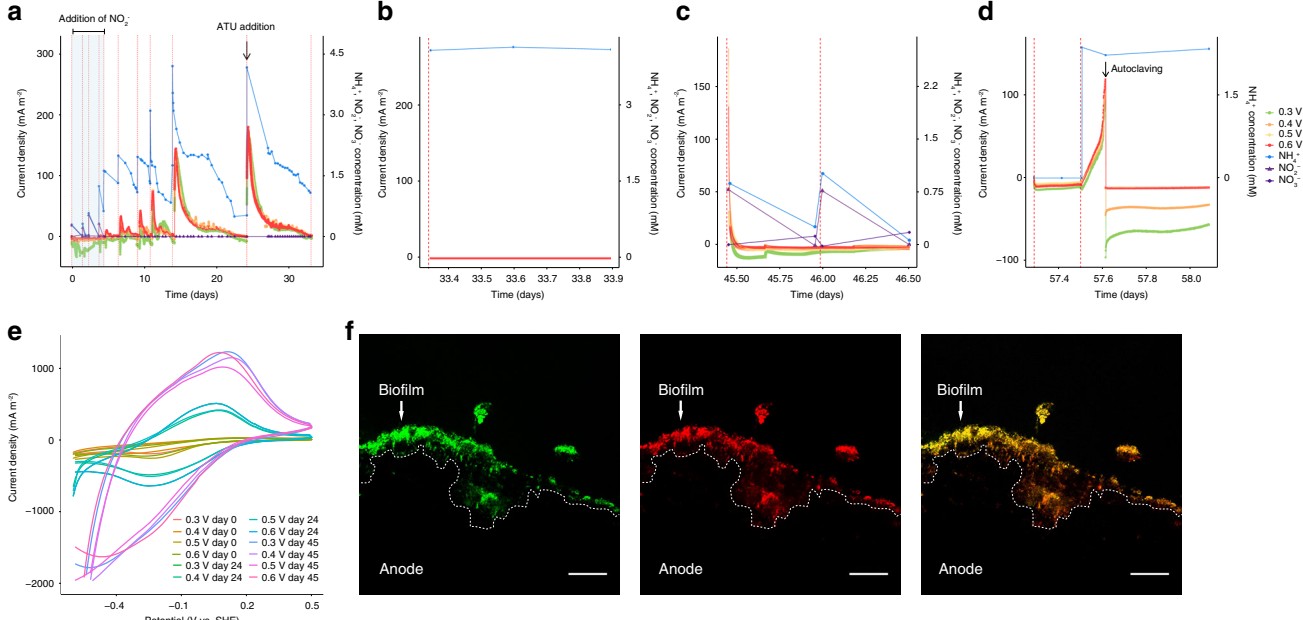

**Fig. 2 Anammox bacteria are electrochemically active. a–d** Ammonium oxidation coupled to current generation in chronoamperometry experiment conducted in one single-chamber multiple working electrode microbial electrolysis cell (MEC) inoculated with *Ca*. Brocadia. **a** MEC was operated initially under different set potentials with the addition of nitrite, which is the preferred electron acceptor for ammonium oxidation by anammox bacteria, followed by operation with working electrodes as sole electron acceptors. The highlighted area in blue refers to the operation of MEC in the presence of nitrite. The black arrow indicates the addition of allylthiourea (ATU), a compound that selectively inhibits nitrifiers. **b** MEC operated under open circuit voltage (OCV) mode. **c** MEC operated at different set potentials and with the addition of nitrite. **d** MEC operated at different set potentials and without the addition of ammonium and then with the addition of ammonium followed by autoclaving. The black arrow in (**d**) indicates autoclaving followed by re-connecting of the MECs. Red dashed lines in (**a–d**) represent a change of batch. **e** Cyclic Voltammogram (1 mV s$^{-1}$) of anammox biofilm grown on working electrodes (i.e., anodes) operated at different set potentials and growth periods following inoculation in MEC. **f** Confocal laser scanning microscopy images of a thin cross-section of the graphite rod anodes (0.6 V vs. standard hydrogen electrode (SHE) applied potential). The images are showing the in-situ spatial organization of all bacteria (green), anammox bacteria (red), and the merged micrograph (yellow). Fluorescence in-situ hybridization was performed with EUB I, II, and III probes for all bacteria[44,45] and Alexa647-labeled Amx820 probe for anammox bacteria[46,47]. The dotted outline indicates the graphite rod anode surface. The white arrow indicates the biofilm. The scale bars represent 20 μm in length.

a biofilm on the surface of the electrodes (Supplementary Fig. 1i) and current generation coupled to $NH_4^+$ oxidation was observed in the absence of $NO_2^-$ (Fig. 2a). Further, $NO_2^-$ and $NO_3^-$ were below the detection limit at all time points when the working electrode was used as the sole electron acceptor, suggesting that $NO_2^-$ and $NO_3^-$ did not play an apparent role in the process. The magnitude of the current generation was proportional to the $NH_4^+$ concentration (Fig. 2a), and maximum current density was observed at the set potential of 0.6 V vs. SHE. There was no visible biofilm growth and current generation at set potentials ≤0.2 V vs. SHE. To confirm that the electrode-dependent anaerobic oxidation of $NH_4^+$ was catalyzed by anammox bacteria, additional control experiments were conducted in chronological order in the MEC. The presence of allylthiourea (ATU), a compound that selectively inhibits aerobic $NH_3$ oxidation by ammonia monooxygenase (AMO) in ammonia-oxidizing bacteria (AOB), ammonia-oxidizing archaea (AOA), and Comammox[19], did not result in an inhibitory effect on $NH_4^+$ removal and current generation (Fig. 2a). $NH_4^+$ was not oxidized when the MEC was operated in open circuit voltage mode (OCV; the electrode is not used as an electron acceptor) (Fig. 2b), strongly suggesting an electrode-dependent $NH_4^+$ oxidation and that trace amounts of $O_2$ if present, are not responsible for $NH_4^+$ oxidation. Addition of $NO_2^-$ resulted in an immediate drop in current density with simultaneous removal of $NH_4^+$ and $NO_2^-$ and formation of $NO_3^-$, in the expected stoichiometry[18] (Fig. 2c). Repeated addition of $NO_2^-$ resulted in the complete abolishment of the current generation, indicating that anammox bacteria were

solely responsible for current production in the absence of an exogenous electron acceptor. Absence of $NH_4^+$ from the feed resulted in no current generation, and current was immediately resumed when $NH_4^+$ was added again to the feed (Fig. 2d), further supporting the role of anammox bacteria in current generation. These results also indicate that current generation was not catalyzed by electrochemically active heterotrophs, which might utilize organic carbon generated from endogenous decay processes. Autoclaving the MEC immediately stopped current generation and $NH_4^+$ removal (Fig. 2d) indicating that current generation was due to biotic reaction. Similar results were also obtained with MECs operated with *Ca*. Scalindua or *K. stutt-gartiensis* cultures (Supplementary Fig. 3a, b), suggesting that they are also electrochemically active and can oxidize $NH_4^+$ using working electrodes as the electron acceptor. Taken together, these results provide strong evidence for electrode-dependent anaerobic oxidation of $NH_4^+$ by phylogenetically distant anammox bacteria. Cyclic voltammetry (CV) was used to correlate between current density and biofilm age, in cell-free filtrates (filtered reactor solution) and the developed biofilms at different time intervals. The anodes exhibited similar redox peaks with midpoint potentials ($E_{1/2}$) of $-0.01 \pm 0.05$ V vs. SHE for all three anammox species (Fig. 2e and Supplementary Fig. 3c, d). The midpoint potentials obtained in our CV analyses were in the redox windows of cytochromes involved in external electron transport in *Shewanella* spp. such as CymA and MtrC[20]. In contrast, our results differ from a previous study that reported the complete anoxic conversion of $NH_4^+$ to $N_2$ at oxidative potentials of

$0.73 \pm 0.06$ V vs. SHE in a nitrifying bioelectrochemical system[21]. This difference in the redox potentials suggest different pathways of anoxic $NH_4^+$ oxidation. No redox peaks were observed for the cell-free solution, indicating that soluble mediators are not involved in EET. Also, the addition of exogenous riboflavin, which is a common soluble mediator involved in flavin-based EET process in gram-positive and gram-negative bacteria[13,22], did not invoke changes in current density. Thus, the CV analysis corroborated that the electrode biofilms were responsible for current generation through direct EET mechanism.

The mole of electrons transferred to the electrode per mole of $NH_4^+$ oxidized to $N_2$ (Supplementary Table 1) was stoichiometrically close to Eq. 1. Also, electron balance calculations showed that coulombic efficiency (CE) was $87.8 \pm 3.2\%$ for all $NH_4^+$ concentrations and anammox cultures tested in the experiments with electrodes as the sole electron acceptor (Supplementary Table 1).

$$2NH_4^+ \xrightarrow{+0.6 \text{ V vs. SHE}} N_2 + 8H^+ + 6e^- \tag{1}$$

To determine if cathodic reaction (i.e., hydrogen evolution reaction) has an effect on electrode-dependent anaerobic $NH_4^+$ oxidation, additional experiments with Ca. Brocadia were conducted by operating single and double-chamber MECs in parallel (at 0.6 V vs. SHE applied potential). However, there was no significant difference in $NH_4^+$ oxidation and current production between the different reactor configurations (Supplementary Fig. 4), suggesting no influence of cathodic reaction (i.e., $H_2$ recycling) on the process. This was further supported by electron balance and CE calculations (Supplementary Table 1). In addition, $NH_4^+$ oxidation and current production were not affected by the addition of Penicillin G (Supplementary Fig. 4), a compound that has inhibitory effects in some heterotrophs, but it does not have any observable short-term effects on anammox activity[23,24]. Similar results were obtained with Ca. Scalindua and K. stuttgartiensis (data not shown). As pointed above, one of the limitations of Penicillin G is that it does not arrest the activity of all heterotrophs. Despite this limitation, the role of heterotrophs in current production was excluded because of the other experimental controls conducted in this study. Since no exogenous organic carbon was added to the MEC reactors, the only source of organics for heterotrophic organisms was through endogenous decay. However, there was no current generation in the absence of $NH_4^+$ (Fig. 2d), suggesting the lack of involvement of heterotrophic electroactive bacteria in the process.

Scanning electron microscopy (SEM) confirmed biofilm formation on the electrodes' surface for the three tested anammox bacteria (Supplementary Fig. 5). The biofilm cell density of MECs inoculated with Ca. Brocadia was higher at 0.6 V vs. SHE (Supplementary Fig. 5e, f) compared with other set potentials, and no biofilm was observed at set potentials ≤0.2 V vs. SHE (Supplementary Fig. 5a). These observations correlate very well with the obtained current profiles at different set potentials (Fig. 2a). Cell appendages between cells and the electrode were not observed. Cell appearance was very similar to reported SEM images of anammox cells[23].

FISH with anammox-specific probes (Fig. 2f) and metagenomics of DNA extracted from the biofilm on the working electrodes of MECs showed that anammox were the most abundant bacteria in the biofilm community (Supplementary Fig. 2b, d). Similarly, AOB were not detected, which further supports the lack of ATU inhibition on $NH_4^+$ removal and current generation. By differential coverage and sequence composition-based binning[25], it was possible to extract high-quality genomes of Brocadia and Scalindua species from the electrodes (Supplementary Fig. 2b, d). Based on the differences in the genome content, average amino acid identity (AAI) ≤ 95% compared with reported anammox genomes to date, and evolutionary divergence in phylogenomics analysis (Supplementary Fig. 6) we propose a tentative name for Ca. Brocadia present in our MECs: Candidatus Brocadia electricigens (etymology: L. adj. electricigens; electricity generator).

**Molecular mechanism of EET-dependent anammox process.** After confirming through bioelectrochemical analyses that anammox bacteria are electrochemically active, isotope labeling experiments were carried out to better understand how $NH_4^+$ is converted to $N_2$ by anammox bacteria in EET-dependent anammox process. Complete oxidation of $NH_4^+$ to $N_2$ was demonstrated by incubating the MECs with $^{15}NH_4^+$ (4 mM) and $^{14}NO_2^-$ (1 mM). Consistent with expected anammox activity, anammox bacteria consumed first the $^{14}NO_2^-$ resulting in the accumulation of $^{29}N_2$ in the headspace of the MECs. Interestingly, after depletion of available $^{14}NO_2^-$, a steady increase of $^{30}N_2$ was observed with slower activity rates compared with the typical anammox process (Fig. 3a, Supplementary Table 2). These results confirm the GO experiments where $^{30}N_2$ was detected when the three anammox species were incubated with $^{15}NH_4^+$ (Fig. 1c). Gas production was not observed in the abiotic control incubations. In the current model of the anammox reaction (Eq. 2)[4], $NH_4^+$ is converted to $N_2$ with $NO_2^-$ as the terminal electron acceptor. This is a process in which first, $NO_2^-$ is reduced to nitric oxide (NO, Eq. 3) and subsequently condensed with ammonia ($NH_3$) to produce hydrazine ($N_2H_4$, Eq. 4), which is finally oxidized to $N_2$ (Eq. 5). The four low-potential electrons released during $N_2H_4$ oxidation fuel the reduction reactions (Eqs. 3 and 4), and are proposed to build up the membrane potential and establish a proton-motive force across the anammoxosome membrane driving the ATP synthesis.

$$NH_4^+ + NO_2^- \rightarrow N_2 + 2H_2O \, (\Delta G^{0\prime} = -357 \text{ kJ mol}^{-1}) \tag{2}$$

$$NO_2^- + 2H^+ + e^- \rightarrow NO + H_2O \, (E^{0\prime} = +0.38 \text{ V}) \tag{3}$$

$$NO + NH_4^+ + 2H^+ + 3e^- \rightarrow N_2H_4 + H_2O \, (E^{0\prime} = +0.06 \text{ V}) \tag{4}$$

$$N_2H_4 \rightarrow N_2 + 4H^+ + 4e^- \, (E^{0\prime} = -0.75 \text{ V}) \tag{5}$$

In the MEC experiments with Ca. Brocadia using multiple working electrodes as sole electron acceptors, we observed the production of $NH_2OH$ followed by a transient accumulation of $N_2H_4$ (Supplementary Fig. 7). No inhibitory effect was observed in incubations with 2-phenyl-4,4,5,5,-tetramethylimidazoline-1-oxyl-3-oxide (PTIO) (Supplementary Fig. 8), a NO scavenger[4]. Therefore, we hypothesized that $NH_2OH$, and not NO, is an intermediate of the electrode-dependent anammox process. To investigate whether $NH_2OH$ could be produced directly from $NH_4^+$ in electrode-dependent anammox process, MECs were incubated with $^{15}NH_4^+$ (4 mM) and $^{14}NH_2OH$ (2 mM). The isotopic composition of the reactors revealed that unlabeled $^{14}NH_2OH$ was used as a pool substrate, and we detected newly synthetized $^{15}NH_2OH$ from $^{15}NH_4^+$ oxidation (Fig. 3b). Similarly, a previous study showed that $NH_2OH$ was the major intermediate of anoxic $NH_4^+$ oxidation performed by electroactive nitrifying microorganisms[21]. Even though Vilajeliu et al.[21] observed the same intermediate, the difference in the community composition and midpoint redox potentials, suggest different pathways of microbial-driven anoxic $NH_4^+$ oxidation to $NH_2OH$. It is known that NO and $NH_2OH$, the known intermediates in the anammox process, are strong competitive inhibitors of the $N_2H_4$

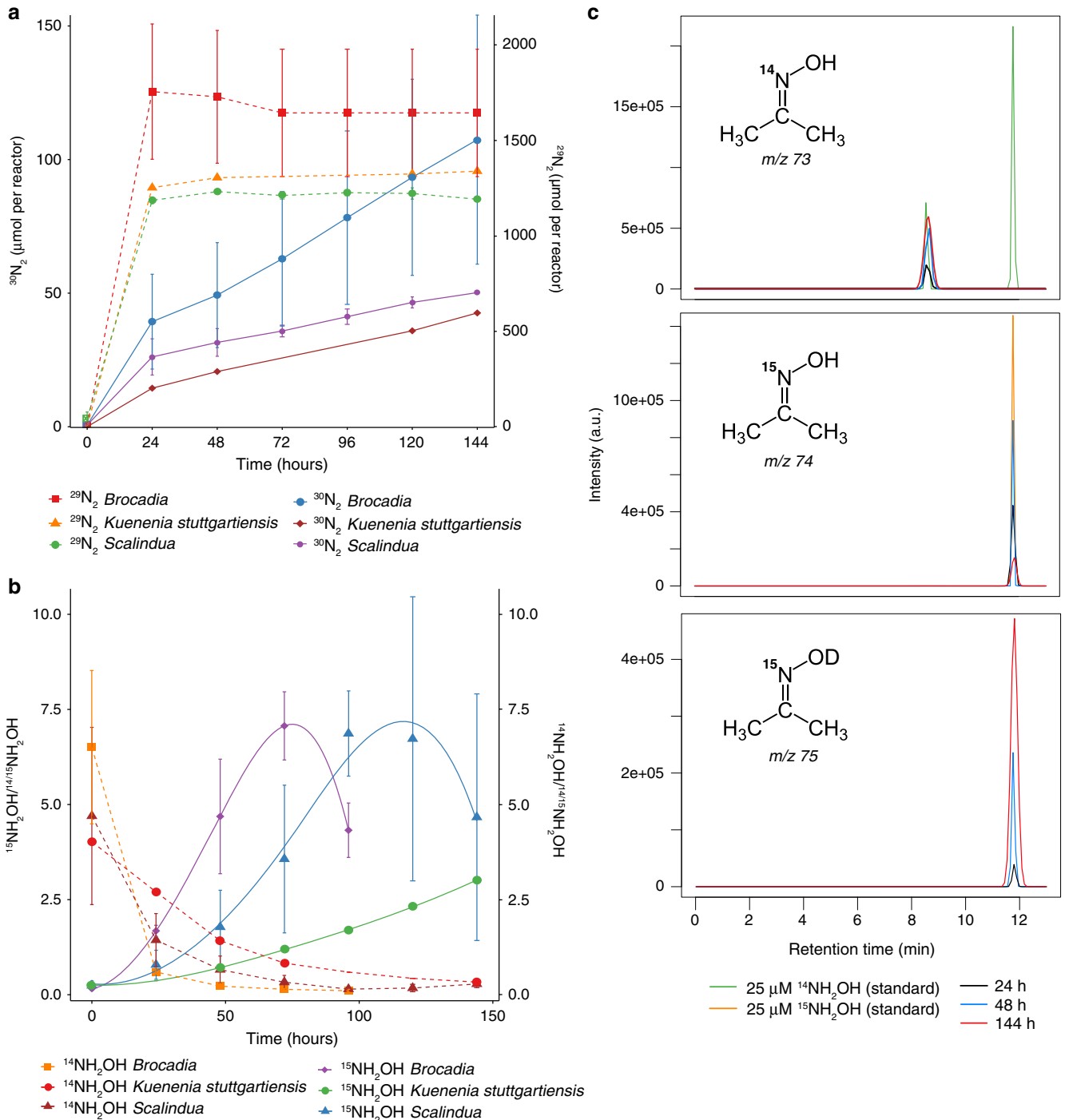

**Fig. 3 Mechanism of NH$_4^+$ oxidation with extracellular electron transfer. a** Time course of the anaerobic oxidation of $^{15}$NH$_4^+$ to $^{29}$N$_2$ and $^{30}$N$_2$. The single-chamber microbial electrolysis cells (MECs) with mature biofilm on the working electrodes operated at 0.6 V vs. SHE were fed with 4 mM $^{15}$NH$_4^+$ and 1 mM $^{14}$NO$_2^-$. Under these conditions, anammox bacteria will consume first the preferred electron acceptor (i.e., $^{14}$NO$_2^-$) and form $^{29}$N$_2$ and then the remaining $^{15}$NH$_4^+$ will be oxidized to the final product ($^{30}$N$_2$) through the electrode-dependent anammox process. NO and N$_2$O were not detected throughout the experiment. Results from triplicate MEC reactors are presented as mean ± SD. **b** Determination of NH$_2$OH as the intermediate of the electrode-dependent anammox process. The MECs with mature biofilm on the working electrodes operated at 0.6 V vs. SHE were fed with 4 mM $^{15}$NH$_4^+$ and 2 mM $^{14}$NH$_2$OH. Under these conditions, anammox bacteria would preferentially consume the unlabeled pool of hydroxylamine (i.e., $^{14}$NH$_2$OH), leading to the accumulation of $^{15}$NH$_2$OH due to the oxidation of $^{15}$NH$_4^+$. Samples were derivatized using acetone, and isotopic ratios were determined by gas chromatography mass spectrometry (GC/MS). Results from triplicate MEC reactors are presented as mean ± SD. **c** Ion mass chromatograms of hydroxylamine derivatization with acetone. The MECs with mature biofilm (*Ca.* Brocadia) on the working electrodes operated at 0.6 V vs. standard hydrogen electrode (SHE) were fed with 4 mM $^{15}$NH$_4^+$ and 10% deuterium oxide (D$_2$O). The mass to charge (m/z) of 73, 74, and 75 corresponds to derivatization products of $^{14}$NH$_2$OH, $^{15}$NH$_2$OH, and $^{15}$NH$_2$OD, respectively, with acetone determined by GC/MS. Twenty microliters of $^{14}$NH$_2$OH and $^{15}$NH$_2$OH were used as standards. The 73 m/z (top) at a retention time of 8.6 min arises from the acetone used for derivatization. The 75 m/z (bottom) accumulation over the course of the experiment indicates that the oxygen used in the anaerobic oxidation of ammonium originates from OH$^-$ of the water molecule.

oxidation activity by the hydrazine dehydrogenase (HDH)[26]. However, oxidation of $N_2H_4$ (Supplementary Fig. 7) and detection of $^{30}N_2$ (Fig. 3a) in our experiments, suggest that even though there might be some inhibition caused by the $NH_2OH$, the HDH is still active. Also, comparative transcriptomics analysis of the electrode's biofilm revealed that the HDH was one of the most upregulated genes when the electrode was used as the electron acceptor instead of $NO_2^-$ (Supplementary discussion). Incubations with $^{15}NH_4^+$ (4 mM) in 10% deuterium oxide ($D_2O$) showed accumulation of $^{15}NH_2OD$, which suggests that in order to oxidize the $NH_4^+$ to $NH_2OH$, the different anammox bacteria use $OH^-$ ions generated from water (Fig. 3c). Abiotic incubations did not show any production of $NH_2OH$ or $NH_2OD$. Based on these results we propose the following reactions for EET-dependent anammox process:

$$2NH_4^+ \xrightarrow{+0.6\,V\,vs\,SHE} N_2 + 8H^+ + 6e^- \tag{1}$$

$$NH_4^+ + H_2O \rightarrow NH_2OH + 3H^+ + 2e^- \tag{6}$$

$$NH_4^+ + NH_2OH \rightarrow N_2H_4 + H_2O + H^+ \tag{7}$$

$$N_2H_4 \rightarrow N_2 + 4H^+ + 4e^- \tag{8}$$

The complete $NH_4^+$ oxidation to $N_2$ coupled with reproducible current production can only be explained by electron transfer from the anammoxosome compartment (energetic central of anammox cells and where the $NH_4^+$ is oxidized) to the electrode. In order to identify the possible pathways involved in $NH_4^+$ oxidation and electron flow through compartments (anammoxosome) and membranes (cytoplasm and periplasm) in EET-dependent anammox process (electrode poised at 0.6 V vs. SHE as electron acceptor) vs. typical anammox process (i.e., nitrite used as electron acceptor), we conducted a genome-centric comparative transcriptomics analysis (Supplementary discussion). Even though comparative transcriptomics analysis is a useful approach for exploring and detecting genes not previously known to play a role in adaptive responses to environmental changes, the levels of mRNA are not directly proportional to the expression level of the protein they encode, and therefore it is difficult to predict protein function and activity from quantitative transcriptome data. Accordingly, the results presented below from the comparative transcriptomics analysis should be read as hypothetical. In the anammoxosome compartment, the genes encoding for ammonium transporter (AmtB), hydroxylamine oxidoreductase and HDH were the most upregulated in response to the electrode as the electron acceptor (Supplementary Table 8). This observation agrees with the $NH_4^+$ removal and oxidation to $N_2$ observed in the MECs and isotope labeling experiments (Figs. 2a and 3a). The genes encoding for NO and $NO_2^-$ reductases (*nir* genes) and their redox couples were significantly downregulated when the electrode was used as the electron acceptor (Supplementary Table 8). This is expected as $NO_2^-$ was not added in the electrode-dependent anammox process. Also, this supports the hypothesis that NO is not an intermediate of the electrode-dependent anammox process and that there was no effect of PTIO when $NO_2^-$ was replaced by the electrode as electron acceptor (Supplementary Fig. 8). Isotope labeling experiments revealed that $NH_2OH$ was the intermediate in EET-dependent anammox process and NO was not detected throughout the experiment (Fig. 3b), suggesting that the production of $NH_2OH$ was not through NO reduction. This was further supported by the observation that the electron transfer module (ETM) and its redox partner whose function is to provide electrons to the hydrazine synthase (HZS) for NO reduction to $NH_2OH$ were downregulated (Supplementary

Table 6). Interestingly, our analysis revealed that the electrons released from the $N_2H_4$ oxidation (Eq. 8) are transferred to the electrode via an EET pathway that is analog to the ones present in metal-reducing organisms such as *Geobacter* spp. and *Shewanella* spp. (Supplementary Fig. 10, Supplementary discussion). Highly expressed cytoplasmic electron carriers such as NADH and ferredoxins can be oxidized at the cytoplasmic membrane by the NADH dehydrogenase (NADH-DH) and/or formate dehydrogenase to directly reduce the menaquinone pool inside the cytoplasmic membrane (Supplementary Table 3). An upregulated protein similar to CymA (tetraheme c-type cytochrome) in *Shewanella* would then oxidize the reduced menaquinones, delivering electrons to highly upregulated periplasmic cytochromes shuttles and to a porin-cytochrome complex that spans the outer membrane (Supplementary Fig. 10, Supplementary Table 3). From this complex, electrons could be directly accepted by the insoluble extracellular electron acceptor. Taken together, the results from the comparative transcriptomics analysis suggest an alternative pathway for $NH_4^+$ oxidation coupled to EET when the working electrode is used as electron acceptor compared with $NO_2^-$ as the electron acceptor.

In conclusion, our study provides the first experimental evidence that phylogenetically and physiologically distant anammox bacteria have EET capability and can couple the oxidation of $NH_4^+$ with the transfer of electrons to carbon-based insoluble extracellular electron acceptors. The prevalence of EET-based respiration has been demonstrated using bioelectrochemical systems for both gram-positive and gram-negative bacteria[13,27]. However, compared with reported EET-capable bacteria, to externalize electrons anammox bacteria have to overcome an additional electron transfer barrier: the anammoxosome compartment. Electrochemically active bacteria are typically found in environments devoid of oxygen or other soluble electron acceptors[27]. Our results show a novel process of anaerobic ammonium oxidation coupled to EET-based respiration of carbon-based insoluble extracellular electron acceptor by both freshwater and marine anammox bacteria and suggest that this process may also occur in natural anoxic environments where soluble electron acceptors are not available. In environments such as anoxic sediments, microbial metabolism is limited by the diffusive supply of electron acceptors[28]. Nitrogen loss by anammox and denitrification are expected to be limited by the diffusive flux of $NO_2^-$ or $NO_3^-$, and/or by $O_2$ diffusion that can be used by aerobic $NH_4^+$ oxidizers. However, $^{30}N_2$ production has been observed in $^{15}NH_4^+$ incubations of sediments at depths below the penetration depth of $NO_2^-$, $NO_3^-$ and $O_2$ in marine[11,28–30] and freshwater[31] environments. These observations cannot be explained by conventional denitrification or anammox process. Interestingly, this phenomenon was observed in sediments rich in metal oxides or natural organic matter such as humic substances[11,31]. EET-dependent anaerobic ammonium oxidation may play a role in such environments devoid of soluble electron acceptors such as $NO_2^-$, $NO_3^-$, and $O_2$. In natural environments, the $NH_4^+$ used by anammox bacteria is derived from heterotrophic pathways of degradation of organic matter[32], and therefore anammox cannot operate independently from the mineralization of organic matter[33]. Ubiquitous high-molecular-weight organic compounds such as humic substances are known to act as terminal electron acceptors in anaerobic microbial respiration[10,34]. Also, humic substances may serve as redox mediators between electron donors and poorly soluble metal oxides minerals in soils and sediments mediating dissimilatory metal oxide reduction[34]. Therefore, humic substances present in natural organic matter may be a good candidate in natural environments of carbon-based electron acceptor for EET-based anaerobic ammonium oxidation, or as redox mediator between

anammox and iron oxides. These results offer a new perspective of a key player involved in the biogeochemical nitrogen cycle, which previously was believed to rely strictly on soluble electron acceptors for $NH_4^+$ oxidation. The fact that anammox bacteria can perform $NH_4^+$ oxidation coupled with EET, suggest that this process may have implications in the global nitrogen cycle by contributing to the nitrogen loss in environments where soluble electron acceptors are unavailable. Therefore, a better understanding of EET processes contributes to our understanding of the cycles that occur on our planet[27]. Also, compared with conventional anammox process, EET-dependent anammox process achieved complete removal of $NH_4^+$ (at low and high concentrations) to nitrogen gas with no accumulation of $NO_2^-$ or $NO_3^-$ or the production of the greenhouse gas $N_2O$. In conventional anammox process (i.e., when $NO_2^-$ is used as the electron acceptor), $NO_3^-$ is generated as a result of the oxidation of $NO_2^-$ by anammox bacteria. Consequently, the effluent from conventional anammox process for wastewater treatment requires further polishing by nitrate reducing organisms before discharge. In contrast, as shown by our results, since $NO_2^-$ was not added in the EET-dependent anammox process, no production of $NO_3^-$ was detected. Given the fact that EET-dependent anammox process can occur at very low voltages (0.3–0.6 V vs. SHE), the process can be powered by renewable energy sources such as wind or solar[35]. Also, the energy released from $NH_4^+$ oxidation can be captured in the form of hydrogen gas. These findings have important implications in energy-efficient treatment of N-rich wastewater using bioelectrochemical systems. Future studies should focus on evaluating the EET-dependent process using anode materials with more conductive surface area such as porous micro-channeled electrodes in order to improve the $NH_4^+$ removal rates.

## Materials and methods

**Enrichment and cultivation of anammox bacteria**. Biomass from upflow column reactors (XK 50/60 Column, GE Healthcare, UK) with *Ca*. Brocadia and *Ca*. Scalindua were harvested and used as inoculum. *Ca*. Brocadia and *Ca*. Scalindua planktonic cells were enriched in two bioreactors (BioFlo®115, New Brunswick, USA) equipped with a microfiltration (average pore size 0.1 μm) hollow fiber membrane module (Zena-membrane, Czech Republic) (Supplementary Fig. 1a). Operating conditions of the membrane bioreactors (MBRs) were described previously[15]. The MBRs were operated at pH 7.5–8.0 and 35 ± 1 °C for *Brocadia* and room temperature (20–25 C) for *Scalindua*. The culture liquid in the MBRs was continuously mixed with a metal propeller at a stirring speed of 150 rpm and purged with 95% Ar–5% $CO_2$ at a flow rate of 10 mL min$^{-1}$ to maintain anaerobic conditions. The inorganic synthetic medium was fed continuously to the reactors at a rate of ~5 L d$^{-1}$ and hydraulic retention time was maintained at one day. The synthetic medium was prepared by adding the following constituents; $NH_4^+$ (2.5–10) mM, $NO_2^-$ (2.5–12) mM, $CaCl_2$ 100 mg L$^{-1}$, $MgSO_4$ 300 mg L$^{-1}$, $KH_2PO_4$ 30 mg L$^{-1}$, $KHCO_3$ 500 mg L$^{-1}$, and trace element solutions[36]. In the case of *Ca*. Scalindua culture, the synthetic medium was prepared using non-sterilized Red Sea water. Samples for microbial community characterization were taken from the MBRs for FISH and metagenomics analysis (see FISH and DNA extraction, metagenome library preparation, sequencing and sequence processing and analysis sections below). A previously enriched *K. stuttgartiensis* culture was also used for the experiments[4].

**Incubation of anammox bacteria with $NH_4^+$ and graphene oxide**. To test whether anammox bacteria have EET capability, the three enriched anammox cultures were incubated in serum vials for 216 h with $^{15}NH_4^+$ and graphene oxide (GO) as a proxy for insoluble extracellular electron acceptor. Standard anaerobic techniques were employed in the batch incubation experiments. All the procedures were performed in the anaerobic chamber (Coy Laboratory Products; Grass Lake Charter Township, MI, USA). Anoxic buffers and solutions were prepared by repeatedly vacuuming and purging helium gas (>99.99%) before experiments. Biomass from the MBRs was centrifuged, washed twice, and suspended in inorganic medium containing 2 mM 4-(2-hydroxyethyl)-1-piperazineethanesulfonic acid (HEPES, pH 7.8) prior to inoculation into the vials. The same composition of the inorganic medium used in the MBRs was supplied to the vials. The cell suspension was dispensed into 100 mL glass serum vials, which were sealed with butyl rubber stoppers and aluminum caps. Biomass concentration in the vials ranged from 0.1 to 0.9 mg–protein mL$^{-1}$. The headspace of the serum vials was replaced by repeatedly vacuuming and purging with pure (>99.99%) helium gas. Positive pressure (50–75 kPa) was added to the headspace to prevent unintentional contamination with ambient air during the incubation and gas sampling. Prior the addition of $^{15}NH_4^+$, the vials were pre-incubated overnight at room temperature (~25 °C) to remove any trace amounts of substrates and oxygen. The activity test was initiated by adding 4 mM $^{15}NH_4Cl$ (Cambridge Isotope Laboratories) and GO to a final concentration of 200 mg L$^{-1}$ using a gas–tight syringe (VICI; Baton Rouge, LA, USA). No $NO_2^-$ or $NO_3^-$ was added to the incubations. The vials were incubated in triplicates at 30 °C for *Ca*. Brocadia and *K. stuttgartiensis* cultures and at room temperature (~25 °C) for vials with *Ca*. Scalindua. Vials without biomass were also prepared as abiotic controls. The concentrations of $^{28}N_2$, $^{29}N_2$, and $^{30}N_2$ gas were determined by gas chromatography mass spectrometry (GC/MS) analysis[37]. Fifty microliters of headspace gas was collected using a gas-tight syringe (VICI; Baton Rouge, LA, USA) and immediately injected into a GC (Agilent 7890A system equipped with a CP-7348 PoraBond Q column) combined with 5975C quadrupole inert MS (Agilent Technologies; Santa Clara, CA, USA), and mass to charge (m/z) = 28, 29, and 30 was monitored. Standard calibration curve of $N_2$ gas was prepared with $^{30}N_2$ standard gas (>98% purity) (Cambridge Isotope Laboratories; Tewksbury, MA, USA). At the end of the batch incubations, DNA was extracted and sequenced for metagenomics analysis (see DNA extraction, metagenome library preparation, sequencing and sequence processing, and analysis section below). To confirm the reduction of the GO, the samples were centrifuged and subjected to dehydration process with absolute ethanol. Samples were maintained in a desiccator until Raman spectroscopy analysis. Raman spectroscopy (StellarNet Inc) was performed with the following settings: Laser 473 nm, acquisition time 20 s, accumulation 5 and objective 50×.

**Bioelectrochemical analyses**. To evaluate if anammox bacteria (*Ca*. Brocadia and *Ca*. Scalindua) are electrochemically active, single-chamber multiple working electrode glass reactors with 500 mL working volume were operated in microbial electrolysis cell (MEC) mode. The working electrodes (anodes) were graphite rods of 8 cm length (7.5 cm inside the reactor) and 0.5 cm in diameter. Platinum mesh was used as counter electrode (cathode) and Ag/AgCl as reference electrode (Bioanalytical Systems, Inc.). A schematic representation of the multiple working electrode MEC is presented in Supplementary Fig. 1h. The multiple working electrodes were operated at a set potential of –0.1, 0, 0.1, 0.2, 0.3, 0.4, 0.5, and 0.6 V vs. SHE. The amperometric current was monitored continuously using a VMP3 potentiostat (BioLogic Science Instruments, USA), with measurements every 60 s and analyzed using EC-lab V 10.02 software. To evaluate if *K. stuttgartiensis* is electrochemically active, experiments were conducted in single-chamber MECs (300 mL working volume) with carbon cloth working electrode (0.6 V vs. SHE). The reactors and working and counter electrodes were sterilized by autoclaving prior to the start of the experiments. The reference electrodes were sterilized by soaking in 3 M NaCl overnight and rinsing with sterile medium. After the reactors were assembled, epoxy glue was used to seal every opening in the reactor to avoid leakage. Gas bags (0.1L Cali -5 -Bond. Calibrate, Inc.) were connected to the MECs to collect any gas generated. The gas composition in the gas bags was analyzed using a gas chromatograph (SRI 8610C gas chromatograph, SRI Instruments).

The inorganic medium composition in the MECs was the same as the one supplied in the MBRs (see enrichment and cultivation of anammox bacteria section above), with variations in the $NH_4^+$ and/or $NO_2^-$ concentration. After preparation, the inorganic medium was boiled, sparged with $N_2$:$CO_2$ (80:20) gas mix for 30 min to remove any dissolved oxygen and finally autoclaved. The autoclaved medium was cooled down to room temperature inside the anaerobic chamber (Coy Laboratory, USA). Prior to the experiments, $KHCO_3$ was weighed in the anaerobic chamber and dissolved in the medium. The reactors were operated in fed-batch mode at 30 °C for *Ca*. Brocadia and *K. stuttgartiensis* cultures and at room temperature (~25 °C) for *Ca*. Scalindua. The medium in the MECs was gently mixed with a magnetic stirrer throughout the course of the experiments. The pH of the MECs was not controlled but was at all times between 7.0 and 7.5. To exclude the effect of abiotic (i.e., non-Faradaic) current, initial operation of the reactors was done without any biomass addition. After biomass inoculation, the MECs were operated with set potentials and optimal conditions for the anammox reaction (i.e., addition of $NH_4^+$ and $NO_2^-$). Afterward, $NO_2^-$ was gradually decreased to 0 mM leaving the working electrodes as the sole electron acceptor. To confirm that the electrode-dependent anaerobic oxidation of $NH_4^+$ was catalyzed by anammox bacteria, additional control experiments were conducted in chronological order including addition of allylthiourea (ATU), operation in open circuit voltage mode (i.e., anodes were not connected to the potentiostat; electrode is not used as electron acceptor), addition of nitrite, operation without addition of $NH_4^+$ and then with addition of $NH_4^+$, and autoclaving. ATU was added to a final concentration of 100 μM to evaluate the contribution of nitrifiers to the process[19]. Biomass from a nitrifying reactor was incubated in triplicate vials with 100 μM of ATU and was used as a positive control for the inhibitory effect of ATU. Throughout the reactor operation, the concentrations of $NH_4^+$, $NO_2^-$, and $NO_3^-$ were determined as described below (see "Analytical methods" section). All experiments were done in triplicate MECs, unless mentioned otherwise.

CV at a scan rate of 1 mV s$^{-1}$ was performed for the anodic biofilms at different time intervals following initial inoculation to determine their redox behavior. Scans ranged from −0.6 to 0.6 V vs. SHE at pH 7.0 and 25 °C. Current was normalized to

the geometric anode surface area. To determine the presence of extracellular secreted redox mediators by anodic communities, CVs were performed with cell-free filtrates (filtered using a 0.2 μm pore diameter filter) collected from the reactors and placed in separate sterile electrochemical cells. Also, experiments were conducted to evaluate the effect of adding riboflavin, which is a common soluble mediator involved in EET in gram-positive and gram-negative bacteria[13,22]. Riboflavin was added to the mature anammox biofilm to a final concentration of 250 nM[22].

To test if cathodic reaction (i.e., hydrogen evolution reaction) has an effect on electrode-dependent anaerobic ammonium oxidation, experiments were also conducted in double-chamber MECs (Supplementary Fig. 1k) with a single carbon cloth working electrode (0.6 V vs. SHE). The anode and cathode chambers in double-chamber MECs were separated by a proton-exchange Nafion membrane. Also, to exclude the effect of heterotrophic activity on the current generation, 500 mg L$^{-1}$ of penicillin G (Sigma-Aldrich, St. Louis, MO)[8] was added in the last batch cycle to inhibit heterotrophs.

To determine the role of NO in the electrode-dependent anammox metabolism, single-chamber MECs were incubated with 4 mM NH$_4^+$ and 100 μM of 2-phenyl-4,4,5,5,-tetramethylimidazoline-1-oxyl-3-oxide (PTIO), a NO scavenger. MECs with 4 mM NH$_4^+$ and without PTIO addition were run in parallel as the negative control. PTIO inhibits *K. stuttgartiensis* activity when NO is an intermediate of the anammox reaction[4], therefore vials with *K. stuttgartiensis* were used as positive control of the effect of PTIO. Liquid samples were taken every day and filtered using a 0.2 μm filter and subjected to determination of NH$_4^+$ concentration as described below (see "Analytical methods" section).

For isotopic and comparative transcriptomics analysis experiments, single-chamber MECs (Adams & Chittenden Scientific Glass, USA) with a single carbon cloth working electrode (0.6 V vs. SHE) and 300 mL working volume were used (Supplementary Fig. 1k).

**$^{15}$N tracer batch experiments in MECs**. To elucidate the molecular mechanism of electrode-dependent anaerobic ammonium oxidation by different anammox bacteria, isotopic labeling experiments were conducted in single-chamber MECs operated at set potential of 0.6 V vs. SHE. All batch incubation experiments were performed in triplicate MECs. MEC incubations without biomass for the $^{15}$N tracer batch experiments were also prepared to exclude any possibility of an abiotic reaction. Standard anaerobic techniques were employed in the batch incubation experiments. All the procedures were performed in the anaerobic chamber (Coy Laboratory Products; Grass Lake Charter Township, MI, USA). Anoxic buffers and solutions were prepared by repeatedly vacuuming and purging helium gas (>99.99%) before the experiments. The purity of $^{15}$N-labeled compounds was greater than 99%. The headspace of the MECs was replaced by repeatedly vacuuming and purging with pure (>99.99%) helium gas. Positive pressure (50–75 kPa) was added to the headspace to prevent unintentional contamination with ambient air during the incubation and gas sampling. Oxidation of NH$_4^+$ to N$_2$ was demonstrated by incubating the MECs with $^{15}$NH$_4$Cl (Cambridge Isotope Laboratories, 4 mM) and $^{14}$NO$_2^-$ (1 mM). The MECs were incubated for 144 h at 30 °C for *Ca.* Brocadia and *K. stuttgartiensis* cultures, and at room temperature (~25 °C) for *Ca.* Scalindua. The concentrations of $^{28}$N$_2$, $^{29}$N$_2$, $^{30}$N$_2$, $^{14}$NO, $^{15}$NO, $^{28}$N$_2$O, $^{29}$N$_2$O, and $^{30}$N$_2$O gas were determined by GC/MS[37]. Fifty microliters of headspace gas was collected using a gas-tight syringe (VICI; Baton Rouge, LA, USA) and immediately injected into a GC (Agilent 7890A system equipped with a CP-7348 PoraBond Q column) combined with 5975C quadrupole inert MS (Agilent Technologies; Santa Clara, CA, USA). Standard calibration curve of N$_2$ gas was prepared with $^{30}$N$_2$ standard gas (>98% purity) (Cambridge Isotope Laboratories; Tewksbury, MA, USA).

To investigate whether hydroxylamine (NH$_2$OH) could be produced directly from NH$_4^+$ in electrode-dependent anaerobic ammonium oxidation by anammox bacteria, single-chamber MECs were incubated with $^{15}$NH$_4$Cl (4 mM, Cambridge Isotope Laboratories) and an unlabeled pool of $^{14}$NH$_2$OH (2 mM) for 144 h. Liquid samples were taken every day and filtered using a 0.2 μm filter and subjected to determination of $^{15}$NH$_2$OH and $^{14}$NH$_2$OH. NH$_2$OH was determined by GC/MS analysis after derivatization using acetone[38]. Briefly, 100 μl of liquid sample was mixed with 4 μl of acetone, and 2 μl of the derivatized sample was injected to a GC (Agilent 7890 A system equipped with a CP-7348 PoraBond Q column) combined with 5975 C quadrupole inert MS (Agilent Technologies; Santa Clara, CA, USA) in splitless mode. NH$_2$OH was derivatized to acetoxime (C$_3$H$_7$NO), and the molecular ion peaks were detected at mass to charge (m/z) = 73 and 74 for $^{14}$NH$_2$OH and $^{15}$NH$_2$OH, respectively. Twenty-five micrometers of $^{14}$NH$_2$OH and $^{15}$NH$_2$OH were used as standards. To determine the source of the oxygen used in the electrode-dependent NH$_4^+$ oxidation to NH$_2$OH, MECs were incubated with $^{15}$NH$_4$Cl (4 mM, Cambridge Isotope Laboratories) in the presence of 10% D$_2$O for 144 h. Stable isotopes of NH$_2$OH were determined by GC/MS analysis after derivatization using acetone as described above.

**Activity and electron balance calculations**. Activities of specific anammox ($^{29}$N$_2$) with nitrite as the preferred electron acceptor and electrode-dependent anammox ($^{30}$N$_2$) with working electrode (0.6 V vs. SHE) as sole electron acceptor were calculated based on the changes in gas concentrations in single-chamber MEC batch

incubations. The activity was normalized against the protein content of the biofilm on the electrodes. Protein content was measured as described below (see "Analytical methods" section).

The moles of electrons recovered as current per mole of NH$_4^+$ oxidized were calculated using

$$n_{CE}\left(NH_4^+\right) = \frac{\int_{t=0}^{t} I\,dt}{NH_4^+ \cdot F}$$

where $I$ is the current (A) obtained from the chronoamperometry, $dt$ (s) is the time interval over which data were collected, NH$_4^+$ is the moles of NH$_4^+$ consumed during the experiment, and $F = 96,485$ C mol$^{-1}$ is Faraday's constant. CE was calculated using

$$CE(\%) = \frac{n_{CE}\left(NH_4^+\right)}{n_{CE\,Theo}\left(NH_4^+\right)} \times 100$$

where $n_{CE\,Theo}(NH_4^+)$ is the theoretical number of moles of electrons (in our case it is three moles of electrons) recovered as current per mole of NH$_4^+$ oxidized.

**Analytical methods**. All samples were filtered through a 0.2 μm pore-size syringe filters (Pall corporation) prior to chemical analysis. NH$_4^+$ concentration was determined photometrically using the indophenol method[39] (lower detection limit = 5 μM). Absorbance at a wavelength of 600 nm was determined using multi-label plate readers (SpectraMax Plus 384; Molecular Devices, CA, USA). NO$_2^-$ concentration was determined by the naphthylethylenediamine method[39] (lower detection limit = 5 μM). Samples were mixed with 4.9 mM naphthylethylenediamine solution, and the absorbance was measured at a wavelength of 540 nm. NO$_3^-$ concentration was measured by HACH kits (HACH, CO, USA; lower detection limit = 0.01 mg l$^{-1}$ NO$_3^-$-N). User's guide was followed for these kits and concentrations were measured by spectrophotometer (D5000, HACH, CO, USA). Concentrations of NH$_2$OH and hydrazine (N$_2$H$_4$) were determined colorimetrically as previously described[40]. For NH$_2$OH, liquid samples were mixed with 8-quinolinol solution (0.48% (w/v) trichloroacetic acid, 0.2% (w/v) 8-hydroxyquinoline and 0.2 M Na$_2$CO$_3$), and heated at 100 °C for 1 min. After cooling down for 15 min, absorbance was measured at 705 nm[41]. N$_2$H$_4$ was derivatized with 2% (w/v) p-dimethylaminobenzaldehyde and absorbance at 460 nm was measured[42]. The concentration of biomass on the working electrodes was determined as protein concentration using the DC Protein Assay Kit (Bio-Rad, Tokyo, Japan) according to manufacturer's instructions. Bovine serum albumin was used as the protein standard.

**Fluorescence in situ hybridization (FISH)**. The microbial community in the MBRs and the spatial distribution of anammox cells on the surface of the graphite rod electrodes was examined by FISH after 30 days of reactor operation. The graphite rod electrodes were cut in the anaerobic chamber with a sterilized tube cutter (Chemglass Life Sciences, US). The electrode samples were fixed with 4% (v/v) paraformaldehyde (PFA), followed by 10 nm cryosectioning at −30 °C (Leica CM3050 S Cryostat). FISH with rRNA-targeted oligonucleotide probes was performed as described elsewhere[43] using the EUB338 probe mix composed of equimolar EUB338 I, EUB338 II, and EUB 338 III[44,45] for the detection of bacteria and probes AMX820 or SCA1309 for anammox[46,47]. Cells were counterstained with 1 μg ml$^{-1}$ DAPI (4′,6-diamidino-2-phenylindole) solution. Fluorescence micrographs were recorded by using a Leica SP7 confocal laser scanning microscope. To determine the relative abundance of anammox bacteria by quantitative FISH, 20 confocal images of FISH probe signals were taken at random locations in each well and analyzed by using the digital image analysis DAIME software as described elsewhere[48].

**Scanning electron microscopy**. The graphite rod electrodes were cut in the anaerobic chamber with sterilized tube cutter (Chemglass Life Sciences, US). The electrode samples were soaked in 2% glutaraldehyde solution containing phosphate buffer (50 mM, pH 7.0) and stored at 4 °C. Sample processing and scanning electron microscopy (SEM) was performed as described elsewhere[49]. Samples from the carbon-cloth electrodes were punched out using a 4.8 mm Ø biopsy punch and placed into a 200 μm cavity of a type A platelet (6 mm diameter; 0.1–0.2 mm depth, Leica Microsystems) and closed with the flat side of a type B platelet (6 mm diameter; 300 μm depth). Platelet sandwiches were cryo-fixed by high-pressure freezing (Leica HPM 100; Leica Microsystems, Vienna, Austria) and stored in liquid nitrogen until use. For Hexamethyldisilazane (HMDS) embedding, frozen samples were freeze-substituted in anhydrous methanol containing 2% osmium tetroxide, 0.2% uranyl acetate, and 1% H$_2$O[50]. The substitution followed several intervals: cells were kept at −90 °C for 47 h; brought to −60 °C at 2 °C per hour and kept at −60 °C for 8 h; brought to −30 °C at 2 °C per hour and kept at −30 °C for 8 h in a freeze-substitution unit (AFS; Leica Microsystems, Vienna, Austria). To remove fixatives the samples were washed four times for 30 min in the AFS device at −30 °C with anhydrous methanol and subsequently infiltrated with HMDS by incubating two times for 15 min with 50% HMDS in anhydrous methanol followed by two times 15 min 100% HMDS. After blotting and air-drying the electrode samples were mounted on specimen stubs using conductive carbon tape and

sputter-coated with gold-palladium before imaging in a JEOL JSM-6335F SEM, operating at 3 kV.

**Metagenomics sequencing and analysis**. Biomass from the vials of the GO experiment was harvested by centrifugation ($4000 \times g$, 42 °C) at the end of the batch incubations. Biofilm samples from the electrodes were collected after 30 days of reactor operation with the working electrode as the sole electron acceptor. The biomass pellet and the electrode samples were suspended in Sodium Phosphate Buffer in the Lysing Matrix E 2 mL tubes (MP Biomedicals, Tokyo, Japan). After 2 min of physical disruption by bead beating (Mini-beadbeater™, Biospec products), the DNA was extracted using the Fast DNA spin kit for soil (MP Biomedicals, Tokyo, Japan) according to the manufacturer's instructions. The DNA was quantified using Qubit (Thermo Fisher Scientific, USA) and fragmented to ~550 bp using a Covaris M220 with microTUBE AFA Fiber screw tubes and the settings: duty factor 20%, peak/displayed power 50 W, cycles/burst 200, duration 45 s, and temperature 20 °C. The fragmented DNA was used for metagenome preparation using the NEB Next Ultra II DNA library preparation kit. The DNA library was paired-end sequenced ($2 \times 301$ bp) on a Hiseq 2500 system (Illumina, USA).

Raw reads obtained in the FASTQ format were processed for quality filtering using Cutadapt package v. 1.10[51] with a minimum phred score of 20 and a minimum length of 150 bp. The trimmed reads were assembled using SPAdes v. 3.7.1[52]. The reads were mapped back to the assembly using minimap2[53] (v. 2.5) to generate coverage files for metagenomic binning. These files were converted to the sequence alignment/map (SAM) format using samtools[54]. Open reading frames (ORFs) were predicted in the assembled scaffolds using Prodigal[55]. A set of 117 hidden Markov models (HMMs) of essential single-copy genes were searched against the ORFs using HMMER3 (http://hmmer.janelia.org/) with default settings, with the exception that option (-cut_tc) was used[56]. Identified proteins were taxonomically classified using BLASTP against the RefSeq protein database with a maximum $e$-value cut-off of $10^{-5}$. MEGAN was used to extract class-level taxonomic assignments from the BLAST output[57]. The script network.pl (http://madsalbertsen.github.io/mmgenome/) was used to obtain paired-end read connections between scaffolds. 16S rRNA genes were identified using BLAST[58] (v. 2.2.28+, and the 16S rRNA fragments were classified using SINA[59] (v. 1.2.11) with default settings except min identity adjusted to 0.80. Additional supporting data for binning was generated according to the description in the mmgenome package[60] (v. 0.7.1). Genome binning was carried out in R[61] (v. 3.3.4) using the R-studio environment. Individual genome bins were extracted using the multimetagenome principles[25] implemented in the mmgenome R package[61] (v. 0.7.1). Completeness and contamination of bins were assessed using coverage plots through the mmgenome R package and by the use of CheckM[62] based on the occurrence of a set of single-copy marker genes[63]. Genome bins were refined manually as described in the mmgenome package and the final bins were annotated using PROKKA[64] (v. 1.12-beta).

**Phylogenomics analysis**. Extracted bins and reported anammox genomes were used for phylogenetic analysis. Reported anammox genomes were downloaded from the NCBI GenBank. HMM profiles for 139 single-copy core genes[63] were concatenated using anvi'o platform[65]. Phylogenetic trees with estimated branch support values were constructed from these concatenated alignments using MEGA7[66] with Neighbor Joining, Maximum-likelihood and UPGMA methods.

**Comparative transcriptomics analysis**. Comparative transcriptomic analysis was conducted to compare the metabolic pathway of $NH_4^+$ oxidation and electron flow when working electrode is used as the electron acceptor vs. $NO_2^-$ as the electron acceptor. Samples for comparative transcriptomic analysis were taken from mature electrode's biofilm of duplicate single-chamber MECs with $NO_2^-$ as the sole electron acceptor and after switching to set potential growth (0.6 V vs. SHE, electrode as the electron acceptor). Biofilm samples were collected from carbon cloth electrodes with sterilized scissors in the anaerobic chamber. Samples were stored in RNAlater™ Stabilization Solution (Invitrogen™) until further processing. Total RNA was extracted from the samples using PowerBiofilm RNA Isolation kit (QiAGEN) according to the manufacturer's instructions. The RNA concentration of all samples was measured in duplicate using the Qubit BR RNA assay. The RNA quality and integrity were confirmed for selected samples using TapeStation with RNA ScreenTape (Agilent Technologies). The samples were depleted of rRNA using the Ribo-zero Magnetic kit (Illumina Inc.) according to manufacturer's instructions. Any potential residual DNA was removed using the DNase MAX kit (MoBio Laboratories Inc.) according to the manufacturer's instructions. After rRNA depletion and DNase treatment the samples were cleaned and concentrated using the RNeasy MinElute Cleanup kit (QIAGEN) and successful rRNA removal was confirmed using TapeStation HS RNA Screentapes (Agilent Technologies). The samples were prepared for sequencing using the TruSeq Stranded Total RNA kit (Illimina Inc.) according to the manufacturer's instructions. Library concentrations were measured using Qubit HS DNA assay and library size was estimated using TapeStation D1000 ScreenTapes (Agilent Technologies). The samples were pooled in equimolar concentrations and sequenced on an Illumina HiSeq2500 using a $1 \times 50$ bp Rapid Run (Illumina Inc).

Raw sequence reads in fastq format were trimmed using USEARCH[67] v10.0.2132, -fastq_filter with the settings -fastq_minlen 45 -fastq_truncqual 20. The trimmed transcriptome reads were also depleted of rRNA using BBDuk[68] with the SILVA database as reference database[69]. The reads were mapped to the predicted protein coding genes generated from Prokka[64] v1.12 using minimap2[53] v2.8-r672, both for the total metagenome and each extracted genome bin. Reads with a sequence identity below 0.98 were discarded from the analysis. The count table was imported to R[61], processed and normalized using the DESeq2 workflow[70] and then visualized using ggplot2. Analyses of overall sample similarity were done using normalized counts (log transformed), through vegan[71] and DESeq2[70] packages (Supplementary Fig. 9). Differentially expressed genes were evaluated for the presence of N-terminal signal sequences, transmembrane spanning helices (TMH) and subcellular localization using SignalP 5.0[72], TMHMM 2.0 software and PSORTb 3.0.2[73] respectively. Differentially expressed genes that appeared annotated as 'hypothetical' were reconsidered for a putative function employing BLAST searches (i.e., BLASTP, CD-search, SmartBLAST), MOTIF search, COG, and PFAM databases, as well as by applying the HHpred homology detection and structure prediction program (MPI Bioinformatics Toolkit).

**Statistics and reproducibility**. The number of replicates is detailed in the subsections for each specific experiment and was mostly determined by the amount of biomass available for the different cultures. In all experiments, three biological replicates were used, unless mentioned otherwise. No statistical methods were used to predetermine the sample size. The experiments were not randomized, and the investigators were not blinded to allocation during experiments and outcome assessment. Statistical analyses were carried out in R[61] v. 3.3.4 using the R-studio environment.

**Reporting summary**. Further information on research design is available in the Nature Research Reporting Summary linked to this article.

## Data availability
Raw sequencing reads of Illumina HiSeq of metagenomics and metatranscriptomics data associated with this project can be found at the NCBI under BioProject PRJNA517785. Annotated GenBank files for the anammox genomes extracted in this study can be found under the accession numbers SHMS00000000 and SHMT00000000. The genome binning and the comparative transcriptomics analysis are entirely reproducible using the R files available on https://github.com/DarioRShaw/Electro-anammox. Also, complete datasets generated in the differential expression analysis are available in the online version of the paper.

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

## Acknowledgements

This work was supported by Center Competitive Funding Program (FCC/1/1971-33-01) to P.E.S. from King Abdullah University of Science and Technology (KAUST). M.S.M.J. was supported by ERC AG 232937 and 339880 and SIAM OCW/NWO 024002002.

## Author contributions

D.R.S. executed the experiments and analyzed the data. M.A. enriched planktonic cells Ca. Brocadia and Scalindua in the MBRs and contributed with the isotope labeling experiments. Bioelectrochemical analysis were done by D.R.S. and K.P.K., R.M., and L.V.N designed and executed the scanning electron microscopy analyses. D.R.S. performed the metagenomics analysis. D.R.S. and M.A. did the phylogenomics analysis. D.R.S. and M.S.M.J. designed the Isotopic batch experiments. D.R.S. and J.R. did the comparative transcriptomics analysis and developed the molecular model. D.R.S., M.A., K.P.K., M.S.M.J., and P.E.S. planned the research. D.R.S. wrote the paper with critical feedback from P.E.S., M.S.M.J., L.V.N, J.A.G, M.A., K.P.K., J.R., and R.M.

## Competing interests

The authors declare no competing interests.
