## [Peer Review File · Nature Communications]

Reviewers' comments:

Reviewer #1 (Remarks to the Author):

I read this article with great pleasure. The results are of great interest and potential significance. It is timely, well written and certainly deserves publication in Nature Communications.

Before a final consideration I do ask the authors to address my following remarks and concerns (chronological order) these are mainly minor, but with one major exception that is a main concern needing revision of the manuscript:

- line 195: The authors state that no other known electroactive microorganisms were found; please specify how this was concluded, as no marker gene does exist. Was this done by comparison to a database?

- line 227 and line 286ff: the authors elaborate that NH_2OH is an important intermediate of the anaerobic NH_4^+ oxidation by anammox, this is convincing, but there is a earlier study by Vilajeliu et al already showing this for a enrichment culture (Water Res. 130 , 168 - 175) please discuss these and also compare the CVs (in line 160ff) you obtained to that reported in this study

- 242 I suggest to report the potential vs. SHE (as this can be straightforward converted into dG also for the non-electrochemist)

- line 635 MAJOR: You report that all experiments were done in triplicate (at least), but this I cannot find standard deviations for the numbers in the text respectively error bars. I see the clear need to calculate and add these to make this very good study as meaningful as it deserves. This holds true for all numbers, e.g. provide CE as $80 \pm X\%$, but especially for the graphs. For instance it is not clear, if Figure 2 a to d shows an example (which I would strongly dislike) or the average (but without an error?)

Please specify and revise.

Reviewer #2 (Remarks to the Author):

Major comments

In this manuscript the authors present interesting results pertaining to extracellular electron transfer during the anaerobic oxidation of ammonia by enrichments of anammox bacteria. An alternate anammox pathway associated with electrogenicity where hydroxylamine (NH_2OH) and not nitric oxide (NO) is presented.

The experiments are largely well designed with controls. The different lines of evidence largely (but not completely) support the very specific and narrow questions posed (are the enrichments containing anammox bacteria capable of external electron transfer). The use of gene expression measurements to infer function are a potential weak link since they only capture part of the information flow. At the very least it would be good to evaluate if the same transcripts have been used to suggest electrogenic activity in other bacteria.

Despite the multi-level investigation of the electrogenic reactors, the authors don't provide convincing evidence that the anammox organisms are the primary or only electroactive bacteria. In this context, a discussion on the role of nitrate is missing. The source of hydroxylamine (assumed to be solely anammox) is also puzzling, given chemical routes for hydroxylamine production. This is the biggest and possibly fatal weakness of this paper – in directly implicating anammox bacteria in electrogenic activity. The work is still good, but the results are not conclusive enough to support the claims made and the novelty is somewhat diluted.

The use of ATU to inhibit aerobic ammonia oxidizing bacteria could be appropriate but using compounds like penicillin to selectively inhibit heterotrophic bacteria is simply unacceptable. The

authors do cite previous work but it is not clear how the penicillin doses were determined for this study or how they compare to previous studies.

The manuscript was not very well written and it was sometimes a struggle to determine what the authors actually meant. In the same vein, some of the citations were not directly linked to the subject matter being discussed.

Minor comments

Different techniques are suggested in different sections of the paper as related to determining the degree of enrichment of anammox bacteria in microcosms. These include fluorescence in-situ hybridization and sequencing. These techniques give different results even when applied to the same sample. It is not clear why multiple or different techniques were used.

L 82-84. While environmental niche differentiation exists amongst different anammox groups, such statements (L82-84) overlook the flexibility amongst microorganisms to respond to their environments. It would be good to acknowledge such flexibility.

L 92-93. Basing reaction occurrence on visual observation is weak. Can GO conversion be chemically determined?

L195-196. The non-detection of known electrogenic bacteria cannot rule out the contribution of hitherto unknown electrogenic bacteria.

The discussion on the reactions inside and outside the anammoxosome are lacking in detail. There is an abrupt mention of it acting as a barrier for electron transfer, without any perspective or further discussion.

Reviewer #3 (Remarks to the Author):

Shaw et al. experimentally show that both multiple distant phylogenetic lineages of anammox bacteria (both marine and freshwater taxa) can couple NH_4^+ oxidation and extracellular electron transfer (EET). This is demonstrated through a number of different experiments that included isotope chemistry, visualization, and sequencing. This work represents a significant advance in the field, working with bacterial organisms that are difficult to culture in highly controlled conditions. The authors also go attempt to better understand pathways that are used by these organisms to overcome the additional barrier of the anammoxosome compartment. While some aspects of the study such as the transcriptomics were not as successful as was likely hoped for, the study nonetheless represents elegant work and an advance in the field. Overall I think that it would be valuable if the author's provided some speculation on how likely it is for this process to occur in natural environments and implications of the occurrence of this reaction in natural environments. The engineering implications are much more well developed. Other minor comments are below.

Specific suggestions are below

Abstract:

Line 26: Who does "their genomes" refer to? Unclear from the sentence structure.

Line 42. What perception is challenged? Make this more clear.

Please provide a more concrete link to bigger picture implications. Why does this coupling matter?

Main Text:

Line 48. Possibly change showed to suggested

Line 50-51. Possibly could shorten this sentence.
However, the mechanism of this coupling reaction has remained unexplored to date.

Line 50. Why would the presence of a coupling of NH_4^+ oxidation and extracellular electron transfer (EET) be important? It would be helpful here to expand on the potential broader significance of this reaction.

Lines 52-63. This section is confusing. It would be helpful to rework to shorten the text and clarify the main points.

Line 303. I would change obvious to another word, such as important

Line 294. How likely is it that this reaction occurs/is prevalent in natural environments? What is an example of a natural anoxic environment where this process might occur and be important?

Grammatical checks:

Some issues with verb tense. Check the text throughout.

e.g.

Line 45. Has to have

Line 63. Is to has been

Line 102. Are to were

Line 117. Use

Also, multiple places are missing articles "a", "an", "the". Check the text throughout.

e.g. Line 39. Add in "an electrode"

Line 82. "an" insoluble

Point-by-point response to Reviewers' comments

Reviewer #1

Note to Reviewer # 1: Our responses to the comments have been divided and numbered to facilitate review. Reviewers' comments appear in *italics*, whereas our responses are non-italicized. All changes in the revised manuscript appear in red.

Comment # 1: I read this article with great pleasure. The results are of great interest and potential significance. It is timely, well written and certainly deserves publication in Nature Communications. Before a final consideration I do ask the authors to address my following remarks and concerns (chronological order) these are mainly minor, but with one major exception that is a main concern needing revision of the manuscript:

Response: we thank the reviewer for noting that the work is of great interest and deserves publication in *Nature Communications*. Also, we appreciate the reviewer for carefully going through our manuscript and providing constructive comments to improve the quality of the manuscript. The following are our responses to the comments.

Comment # 2: line 195: The authors state that no other known electroactive microorganisms were found; please specify how this was concluded, as no marker gene does exist. Was this done by comparison to a database?

Response: The reviewer is referring to a sentence at Line 195 (original manuscript), where we wrote, "Also, no other known electrochemically active bacteria were detected in the metagenomes." We agree with the reviewer that there is no marker gene to detect electroactive microorganisms. Also, there is no database available with known genes/proteins and pathways involved in extracellular electron transfer, and therefore we cannot determine if an organism is electroactive or not by merely using 'omics analyses'. Since no marker gene or database exist for electroactive microorganisms, we wrote that sentence at Line 195 (original manuscript) thinking of known model heterotrophic electroactive bacteria such as *Geobacter* and *Shewanella*. To avoid any confusion, we deleted the below sentences from the revised manuscript.

Line 201: "FISH with anammox-specific probes (Fig. 2f) and metagenomics of DNA extracted from the biofilm on the working electrodes of MECs showed that anammox were the most abundant bacteria in the biofilm community (Supplementary Fig. 2b and d). ~~Also, no other known electrochemically active bacteria were detected in the metagenomes.~~ Similarly, AOB were not detected, which further supports the lack of ATU inhibition on NH_4^+ removal and current generation."

Line 109: "Differential coverage showed that the metagenomes were dominated by anammox bacteria (Supplementary Fig. 2a and c). ~~Also, no known EET-capable bacteria were detected in the metagenomes.~~"

However, we would like to emphasize that in our system, current production was solely due to the electro-activity of anammox bacteria and we reached to this conclusion through several solid experimental evidences already presented and explained in the original manuscript (see points 1 to 4 below).

1) The majority of known electroactive bacteria in the literature are heterotrophs and use organic compounds as electron donor. No exogenous organic carbon was added to our system, and therefore

the only source of organics for heterotrophic electroactive bacteria was endogenous decay. Further, current generation was observed only when NH_4^+ was added to our system. However, our control experiments showed that there was no current generation in the absence of NH_4^+ , as presented in the original manuscript, suggesting the lack of involvement of heterotrophic electroactive bacteria in the process. In the original manuscript we wrote:

Line 152: "Absence of NH_4^+ from the feed resulted in no current generation, and current was immediately resumed when NH_4^+ was added again to the feed (Fig. 2d), further supporting the role of anammox bacteria in current generation. These results also indicate that current generation was not catalyzed by electrochemically active heterotrophs, which might utilize organic carbon generated from endogenous decay processes."

2) The addition of Penicillin G, a compound that has inhibitory effect in some heterotrophs but not on anammox bacteria did not have any effect on NH_4^+ oxidation and current production as presented in the original manuscript:

Line 188: "In addition, NH_4^+ oxidation and current production were not affected by the addition of Penicillin G (Supplementary Fig. 4), a compound that has inhibitory effects in some heterotrophs, but it does not have any observable short-term effects on anammox activity^{23,24}. This further supports that current generation was not catalyzed by electrochemically active heterotrophs."

3) Addition of NO_2^- (preferred electron acceptor of anammox bacteria) to the medium, after the biofilm was formed, stopped the stable current production and NH_4^+ and NO_2^- were removed following the typical anammox stoichiometry, suggesting a clear role of anammox bacteria in the NH_4^+ oxidation and current production. In the original manuscript we wrote:

Line 148: "Addition of NO_2^- resulted in an immediate drop in current density with simultaneous removal of NH_4^+ and NO_2^- and formation of NO_3^- , in the expected stoichiometry¹⁸ (Fig. 2c). Repeated addition of NO_2^- resulted in the complete abolishment of the current generation, indicating that anammox bacteria were solely responsible for current production in the absence of an exogenous electron acceptor."

4) In our study, anammox bacteria were solely responsible for NH_4^+ oxidation because addition of ATU, a compound that selectively inhibit nitrifiers, did not inhibit NH_4^+ oxidation or current generation as described in the original manuscript:

Line 140: "To confirm that the electrode-dependent anaerobic oxidation of NH_4^+ was catalyzed by anammox bacteria, additional control experiments were conducted in chronological order in the MEC. The presence of ATU, a compound that selectively inhibits aerobic NH_3 oxidation by ammonia monooxygenase (AMO) in ammonia-oxidizing bacteria (AOB), ammonia-oxidizing archaea (AOA) and Comammox¹⁹, did not result in an inhibitory effect on NH_4^+ removal and current generation (Fig. 2a)."

Comment # 3: line 227 and line 286ff: the authors elaborate that NH_2OH is an important intermediate of the anaerobic NH_4^+ oxidation by annamox, this is convincing, but there is a earlier study by Vilajeliu et al already showing this for a enrichment culture (Water Res. 130 , 168 - 175) please discuss these and also compare the CVs (in line 160ff) you obtained to that reported in this study.

Response: We appreciate the reviewer acknowledging that NH_2OH as an important intermediate in EET-dependent anammox process. As suggested by the reviewer we discussed the study by Vilajeliu et al and emphasized the role of NH_2OH as an important intermediate of microbial-catalyzed anoxic

ammonium oxidation in bioelectrochemical systems, and we provided a comparison of the CVs between the two studies. We now write:

Line 163: “Cyclic voltammetry (CV) was used to correlate between current density and biofilm age, in cell-free filtrates (filtered reactor solution) and the developed biofilms at different time intervals. The anodes exhibited similar redox peaks with midpoint potentials ($E_{1/2}$) of -0.01 ± 0.05 V vs. SHE for all three anammox species (Fig. 2e and Supplementary Fig. 3c and d). The midpoint potentials obtained in our CV analyses were in the redox windows of cytochromes involved in external electron transport in *Shewanella* spp. such as CymA and MtrC²⁰. In contrast, our results differ from a previous study that reported the complete anoxic conversion of NH_4^+ to N_2 at oxidative potentials of 0.73 ± 0.06 V vs. SHE in a nitrifying bioelectrochemical system²¹. This difference in the redox potentials suggest different pathways of anoxic NH_4^+ oxidation.”

Line 236: “The isotopic composition of the reactors revealed that unlabeled $^{14}\text{NH}_2\text{OH}$ was used as a pool substrate, and we detected newly synthesized $^{15}\text{NH}_2\text{OH}$ from $^{15}\text{NH}_4^+$ oxidation (Fig. 3b). Similarly, a previous study showed that NH_2OH was the major intermediate of anoxic NH_4^+ oxidation performed by electroactive nitrifying microorganisms²¹. Even though Vilajeliu et al.²¹ observed the same intermediate, the difference in the community composition and midpoint redox potentials, suggest different pathways of microbial-driven anoxic NH_4^+ oxidation to NH_2OH .”

Comment # 4: 242 I suggest to report the potential vs. SHE (as this can be straightforward converted into dG also for the non-electrochemist)

Response: We thank the reviewer for this suggestion. We now report all the potentials in terms of standard hydrogen electrode (SHE) in the revised manuscript text, figures and supplementary files.

Comment # 5: line 635 MAJOR: You report that all experiments were done in triplicate (at least), but this I cannot find standard deviations for the numbers in the text respectively error bars. I see the clear need to calculate and add these to make this very good study as meaningful as it deserves. This holds true for all numbers, e.g. provide CE as $80 \pm X\%$, but especially for the graphs. For instance, it is not clear, if Figure 2 a to d shows an example (which I would strongly dislike) or the average (but without an error?) Please specify and revise.

Response: We thank the reviewer for this valuable comment. Also, we agree with the reviewer that we should report the standard deviations in the text and error bars in the figures. It should be noted that in the original materials and methods we mentioned that all experiments were done in triplicates, unless mentioned otherwise. For example, we wrote:

Line 437: “All experiments were done in triplicate MECs, unless mentioned otherwise.”

Line 661: “In all experiments, three biological replicates were used, unless mentioned otherwise.”

For the experiments that were done in triplicates, the error bars were already shown in the figures. Also, we now added the standard deviations in the text. We now write:

Line 178: “Also, electron balance calculations showed that coulombic efficiency (CE) was $87.9 \pm 3.2\%$ for all NH_4^+ concentrations and anammox cultures tested in the experiments with electrodes as the sole electron acceptor (Supplementary Table 1).”

As for reviewer's comment with regards Figure 2a to d, we initially tested eight different potentials in parallel (ranging from -0.1 to 0.6 V vs standard hydrogen electrode; SHE) using one single-chamber MEC with multiple working electrodes because we did not know at which applied potential anoxic removal of ammonium will occur. Even though the experiment was done with a single reactor, we performed the same electrochemical evaluation with three different species of anammox bacteria. Figure 2 a to d show the operation of the single MEC reactor with *Ca. Brocadia*. The results obtained with the other two species (i.e., *Ca. Scalindua* and *K. stuttgartiensis*) are presented in Supplementary Figure 3. The three tested species showed similar electrochemical behavior. After determining the optimal applied potential (i.e., 0.6 V vs. SHE), the rest of the experiments were conducted in replicate single-chamber MEC reactors and the standard deviations were presented in the figures and tables.

In order to avoid confusion, we now specify in Figure 2 caption that the results are for one single-chamber multiple working electrode MEC:

Line 889: "Fig. 2 *Ca. Brocadia* is electrochemically active (i.e., able to release electrons from inside the cell to working electrode). (a-d) Ammonium oxidation coupled to current generation in chronoamperometry experiment conducted in **one** single-chamber multiple working electrode MEC inoculated with *Ca. Brocadia*."

Reviewer #2

Note to Reviewer # 2: Our responses to the comments have been divided and numbered to facilitate review. Reviewers' comments appear in *italics*, whereas our responses are non-italicized. All changes in the revised manuscript appear in red.

Comment # 1: In this manuscript the authors present interesting results pertaining to extracellular electron transfer during the anaerobic oxidation of ammonia by enrichments of anammox bacteria. An alternate anammox pathway associated with electrogenicity where hydroxylamine (NH₂OH) and not nitric oxide (NO) is presented.

Response: We thank the reviewer for acknowledging our work and the novel anammox pathway presented in our study. Also, we thank the reviewer for carefully going through our manuscript and providing constructive comments to improve the quality of the manuscript. The following are our responses to the comments.

Comment # 2: The experiments are largely well designed with controls. The different lines of evidence largely (but not completely) support the very specific and narrow questions posed (are the enrichments containing anammox bacteria capable of external electron transfer). The use of gene expression measurements to infer function are a potential weak link since they only capture part of the information flow. At the very least it would be good to evaluate if the same transcripts have been used to suggest electrogenic activity in other bacteria. Despite the multi-level investigation of the electrogenic reactors, the authors don't provide convincing evidence that the anammox organisms are the primary or only electroactive bacteria.

Response: We thank the reviewer for noting that the experiments are well designed with controls. We also agree with the reviewer that just using transcriptomics analysis it is not possible to infer function (in this case electrogenic activity). However, we would like to point out that we did not mention anywhere in the manuscript that gene expression was used to infer electro-activity of anammox bacteria. Transcriptomics analysis was done mainly to compare the metabolic pathway of NH₄⁺ oxidation and electron flow when working electrode is used as electron acceptor versus NO₂⁻ as electron acceptor, as written in the original version of the manuscript:

Line 258: "In order to compare the pathway of NH₄⁺ oxidation and electron flow through compartments (anammoxosome) and membranes (cytoplasm and periplasm) in EET-dependent anammox process (electrode poised at 0.6 V vs. SHE as electron acceptor) versus typical anammox process (i.e., nitrite used as electron acceptor), we conducted a genome-centric comparative transcriptomics analysis (Supplementary discussion)."

The reviewer suggested to evaluate if the same transcripts have been used to suggest electrogenic activity in other bacteria. However, as mentioned by Reviewer #1 and in our response to Comment #2 by Reviewer # 1, there is no marker gene to detect electroactive microorganisms and their activity. Also, there is no database with known genes/proteins and pathways involved in extracellular electron transfer, and therefore it is not possible through merely metagenomics or metatranscriptomics analyses to determine if an organism is electroactive or not. We reached to our conclusion about electrogenic activity of anammox in the electrochemical reactors based on bioelectrochemistry and stable isotope analyses as explained below.

We respectfully disagree with the reviewer's comment "the authors don't provide convincing evidence that the anammox organisms are the primary or only electroactive bacteria". We would like to

emphasize that in our system, anammox bacteria were solely responsible for NH_4^+ oxidation and current generation and that other autotrophic or heterotrophic electroactive bacteria (known or unknown) were not involved in current generation. This conclusion is based on solid experimental evidences (see evidence 1 to 6 below) that we provided in the original manuscript.

1) In our study, anammox bacteria were solely responsible for NH_4^+ oxidation because addition of ATU, a compound that selectively inhibit nitrifiers, did not inhibit NH_4^+ oxidation or current generation as described in the original manuscript:

Line 140: “To confirm that the electrode-dependent anaerobic oxidation of NH_4^+ was catalyzed by anammox bacteria, additional control experiments were conducted in chronological order in the MEC. The presence of ATU, a compound that selectively inhibits aerobic NH_3 oxidation by ammonia monooxygenase (AMO) in ammonia-oxidizing bacteria (AOB), ammonia-oxidizing archaea (AOA) and Comammox¹⁹, did not result in an inhibitory effect on NH_4^+ removal and current generation (Fig. 2a).”

2) Addition of NO_2^- (preferred electron acceptor of anammox bacteria) to the medium, after the biofilm was formed, stopped the stable current production and NH_4^+ and NO_2^- were removed following the typical anammox stoichiometry, suggesting a clear role of anammox bacteria in the NH_4^+ oxidation and current production. In the original manuscript we wrote:

Line 148: “Addition of NO_2^- resulted in an immediate drop in current density with simultaneous removal of NH_4^+ and NO_2^- and formation of NO_3^- , in the expected stoichiometry¹⁸ (Fig. 2c). Repeated addition of NO_2^- resulted in the complete abolishment of the current generation, indicating that anammox bacteria were solely responsible for current production in the absence of an exogenous electron acceptor.”

3) The majority of known electroactive bacteria in the literature are heterotrophs and use organic compounds as electron donor. No exogenous organic carbon was added to our system, and therefore the only source of organics for heterotrophic electroactive bacteria was endogenous decay. Further, current generation was observed only when NH_4^+ was added to our system. However, our control experiments showed that that there was no current generation in the absence of NH_4^+ , as presented in the original manuscript, suggesting the lack of involvement of heterotrophic electroactive bacteria in the process. In the original manuscript we wrote:

Line 152: “Absence of NH_4^+ from the feed resulted in no current generation, and current was immediately resumed when NH_4^+ was added again to the feed (Fig. 2d), further supporting the role of anammox bacteria in current generation. These results also indicate that current generation was not catalyzed by electrochemically active heterotrophs, which might utilize organic carbon generated from endogenous decay processes.”

4) The addition of Penicillin G, a compound that has inhibitory effect in some heterotrophs but not on anammox bacteria did not have any effect on NH_4^+ oxidation and current production as presented in the original manuscript:

Line 188: “In addition, NH_4^+ oxidation and current production were not affected by the addition of Penicillin G (Supplementary Fig. 4), a compound that has inhibitory effects in some heterotrophs, but it does not have any observable short-term effects on anammox activity^{23,24}. This further supports that current generation was not catalyzed by electrochemically active heterotrophs.

5) Anammox bacteria were the most abundant organism in the electrode biofilm and cyclic voltammetry analysis (CV) showed that the electrode biofilms were only responsible for current generation. In the original manuscript we wrote:

Line 201: “FISH with anammox-specific probes (Fig. 2f) and metagenomics of DNA extracted from the biofilm on the working electrodes of MECs showed that anammox were the most abundant bacteria in the biofilm community (Supplementary Fig. 2b and d).”

Line 171: “No redox peaks were observed for the cell-free solution, indicating that soluble mediators are not involved in EET. Also, the addition of exogenous riboflavin, which is a common soluble mediator involved in flavin-based EET process in gram-positive and gram-negative bacteria^{13,22}, did not invoke changes in current density. Thus, the CV analysis corroborated that the electrode biofilms were responsible for current generation through direct EET mechanism.”

6) Stable isotope experiments showed that anammox bacteria were responsible for the complete oxidation of $^{15}\text{NH}_4^+$ to $^{30}\text{N}_2$ using the electrode as the electron acceptor just after $^{14}\text{NO}_2^-$ was depleted to $^{29}\text{N}_2$ gas. Please note that $^{14}\text{NO}_2^-$ is the preferred electron acceptor of anammox bacteria and that the formation of $^{29}\text{N}_2$ gas is an exclusive indicator of anammox activity. In the original manuscript we wrote:

Line 215: “Complete oxidation of NH_4^+ to N_2 was demonstrated by incubating the MECs with $^{15}\text{NH}_4^+$ (4 mM) and $^{14}\text{NO}_2^-$ (1 mM). Consistent with expected anammox activity, anammox bacteria consumed first the $^{14}\text{NO}_2^-$ resulting in the accumulation of $^{29}\text{N}_2$ in the headspace of the MECs. Interestingly, after depletion of available $^{14}\text{NO}_2^-$, a steady increase of $^{30}\text{N}_2$ was observed with slower activity rates compared to the typical anammox process (Fig 3a, Supplementary Table 2).”

Comment # 3: In this context, a discussion on the role of nitrate is missing.

Response: We thank the reviewer for his/her concern about the role of nitrate in the EET-dependent anammox process. We would like to bring to the reviewer’s attention that we did not find any significant role of nitrate when the anode was used as the sole electron acceptor as described in the original manuscript:

Line 102: “Further, isotope analysis of the produced N_2 gas showed that anammox cells were capable of $^{30}\text{N}_2$ formation (Fig. 1c). In contrast, $^{29}\text{N}_2$ production was not significant in any of the tested anammox species or controls, suggesting that unlabeled NO_2^- or NO_3^- were not involved.”

Line 132: “When the exogenous electron acceptor (i.e., NO_2^-) was completely removed from the feed, anammox cells began to form a biofilm on the surface of the electrodes (Supplementary Fig. 1i) and current generation coupled to NH_4^+ oxidation was observed in the absence of NO_2^- (Fig. 2a). Further, NO_2^- and NO_3^- were below the detection limit at all time points when the working electrode was used as the sole electron acceptor.”

Line 327: “EET-dependent anammox process achieved complete removal of NH_4^+ (at low and high concentrations) to nitrogen gas with no accumulation of NO_2^- or NO_3^- or the production of the greenhouse gas N_2O .”

Also, in the original supplementary information associated to the manuscript, we discussed the expression levels of nitrite:nitrate oxidoreductases obtained in the differential expression analysis:

Supplementary information Line 51: “In contrast, the genes encoding for NO and NO₂⁻ reductases (*nir* genes) and their redox couples were significantly downregulated (Supplementary Fig. 10, Supplementary Table 8). This agrees, with the fact that NO₂⁻ and NO₃⁻ were below the detection limit in the MECs (Fig. 2a, Supplementary Fig. 3a and b)”

Supplementary information Line 62: “On the other hand, the *nxr* genes encoding for the soluble nitrite:nitrate oxidoreductase maintained similar levels of expression under both conditions (Supplementary Table 9). However, cytochromes of the *nxr* gene cluster and the hypothetical membrane-bound NXR were found downregulated under set-potential (Supplementary Table 8).”

Comment # 4: The source of hydroxylamine (assumed to be solely anammox) is also puzzling, given chemical routes for hydroxylamine production. This is the biggest and possibly fatal weakness of this paper – in directly implicating anammox bacteria in electrogenic activity. The work is still good, but the results are not conclusive enough to support the claims made and the novelty is somewhat diluted.

Response: We appreciate the reviewer’s concern about the possible production of hydroxylamine through abiotic chemical routes. In the sections on “Electroactivity of anammox bacteria” and “Molecular mechanism of EET-dependent anammox process” of the original manuscript, we provided solid experimental evidence that anammox bacteria were solely responsible for current production coupled with ammonium oxidation to hydroxylamine. Also, we would like to point out that the production of hydroxylamine through abiotic chemical routes was excluded in our study because we had appropriate abiotic controls in all the experiments that we conducted. In the original material and methods section, we wrote:

Line 381: “Vials without biomass were also prepared as abiotic controls.”

Line 424: “To exclude the effect of abiotic (i.e., non-Faradaic) current, initial operation of the reactors was done without any biomass addition.”

Line 471: “MEC incubations without biomass for the ¹⁵N tracer batch experiments were also prepared to exclude any possibility of an abiotic reaction.”

Abiotic ammonium oxidation and abiotic formation of hydroxylamine were not observed in our abiotic controls, thus excluding any possible production of hydroxylamine through chemical routes. In the original version of the manuscript we wrote:

Line 106: “Gas production was not observed in the abiotic control (Fig. 1c).”

Line 124: “No current and NH₄⁺ removal were observed in any of the abiotic controls.”

Line 222: “Gas production was not observed in the abiotic control incubations.”

Line 251: “Abiotic incubations did not show any production of NH₂OH or NH₂OD.”

Also, after autoclaving the reactors, ammonium oxidation and production of hydroxylamine was not observed (Fig. 2d and Supplementary Fig. 7), further confirming that hydroxylamine production was solely by anammox bacteria and not chemical routes. In the original manuscript we wrote:

Line 157: “Autoclaving the MECs immediately stopped current generation and NH_4^+ removal (Fig. 2d) indicating that the current generation was due to biotic reaction.”

Comment # 5: The use of ATU to inhibit aerobic ammonia oxidizing bacteria could be appropriate but using compounds like penicillin to selectively inhibit heterotrophic bacteria is simply unacceptable. The authors do cite previous work but it is not clear how the penicillin doses were determined for this study or how they compare to previous studies.

Response: We agree with reviewer that Penicillin G does not inhibit all heterotrophs, and that is why we stated in the original version of the manuscript that Penicillin G has inhibitory effects in some heterotrophs:

Line 188: “In addition, NH_4^+ oxidation and current production were not affected by the addition of Penicillin G (Supplementary Fig. 4), a compound that has inhibitory effects in some heterotrophs, but it does not have any observable short-term effects on anammox activity^{23,24}.”

The penicillin doses used in our study were determined based on the cited previous studies, where they evaluated the effect and inhibitory doses of Penicillin G on anammox bacteria. Penicillin G does have inhibitory effect in long-term (continuous cultivation) incubations, but it does not have any observable short-term (batch incubations) effects on anammox activity as evidenced by Hu et al. (2013). Hu et. al (2013) wrote:

“The effects of these compounds were determined in both short-term batch incubations and long-term (continuous-cultivation) growth experiments in membrane bioreactors. Lysozyme at 1 g/liter (20 mM EDTA) lysed anammox cells in less than 60 min, whereas penicillin G did not have any observable short-term effects on anammox activity. Penicillin G (0.5, 1, and 5 g/liter) reversibly inhibited the growth of anammox bacteria in continuous-culture experiments”.

The experiments in our study with Penicillin G were conducted in batch incubations. Therefore, Penicillin G was appropriate to differentiate heterotrophs from anammox bacteria. Further, this methodology and concentrations have been used in different studies to differentiate heterotrophs from anammox (e.g., in Oshiki, M. *et al* 2013: “... particularly that of “*Ca. Brocadia sinica*,” were carefully evaluated by using highly enriched cultures and supplementation with antibiotics (penicillin G and chloramphenicol) that are not active against anammox bacteria but inhibit the activity of most heterotrophs (42, 43).”

Most importantly, we would like to emphasize that the role of heterotrophic electroactive organisms in current production was excluded not only because of the tests with Penicillin G, but also because of our experimental controls. The majority of known electroactive bacteria in the literature are heterotrophs and use organic compounds as electron donor. No exogenous organic carbon was added to our system, and therefore the only source of organics for heterotrophic electroactive bacteria was endogenous decay. Further, current generation was observed only when NH_4^+ was added to our system. However, our control experiments showed that that there was no current generation in the absence of NH_4^+ , as presented in the original manuscript, suggesting the lack of involvement of heterotrophic electroactive bacteria in the process. In the original manuscript we wrote:

Line 152: “Absence of NH_4^+ from the feed resulted in no current generation, and current was immediately resumed when NH_4^+ was added again to the feed (Fig. 2d), further supporting the role of anammox bacteria in current generation. These results also indicate that current generation was not catalyzed by electrochemically active heterotrophs, which might utilize organic carbon generated from endogenous decay processes.”

Comment # 6: The manuscript was not very well written and it was sometimes a struggle to determine what the authors actually meant. In the same vein, some of the citations were not directly linked to the subject matter being discussed.

Response: We respect the reviewer's comment, which disagrees with Reviewer #1 who wrote: "I read this article with great pleasure. The results are of great interest and potential significance. It is timely, well written and certainly deserves publication in Nature Communications". Also, Reviewer # 3 did not mention in any of the comments that the manuscript was not well written. Further, we believe that all the references cited in the manuscript were directly linked and relevant to the discussion. Since the reviewer did not provide any example to support "The manuscript was not very well written" and "some of the citations were not directly linked to the subject matter being discussed", no change was made regarding comment #6.

Comment # 7: Different techniques are suggested in different sections of the paper as related to determining the degree of enrichment of anammox bacteria in microcosms. These include fluorescence in-situ hybridization and sequencing. These techniques give different results even when applied to the same sample. It is not clear why multiple or different techniques were used.

Response: Please note that in our study both metagenomics and FISH provided similar results that anammox were the most abundant bacteria in the MBR enrichments and the electrodes' biofilm as mentioned in the original manuscript:

Line 201: "FISH with anammox-specific probes (Fig. 2f) and metagenomics of DNA extracted from the biofilm on the working electrodes of MECs showed that anammox were the most abundant bacteria in the biofilm community (Supplementary Fig. 2b and d)."

We would like to bring to the reviewer's attention that FISH was not only used in this study to determine the relative abundance of anammox bacteria but also to determine the biofilm thickness and the spatial distribution of the biofilm on the electrodes (see Fig. 2f), which cannot be provided by metagenomics analysis. We explained this rationale in the materials and methods section of the original manuscript:

Line 540: "The microbial community in the MBRs and the spatial distribution of anammox cells on the surface of the graphite rod electrodes was examined by FISH after 30 days of reactor operation"

Comment # 8: L 82-84. While environmental niche differentiation exists amongst different anammox groups, such statements (L82-84) overlook the flexibility amongst microorganisms to respond to their environments. It would be good to acknowledge such flexibility.

Response: We thank the reviewer for this valuable suggestion. We agree that anammox bacteria are versatile and can adapt to a variety of environments. We now write in the revised manuscript:

Line 87: "Cultures of *Ca. Brocadia* (predominantly adapted to freshwater environments) and *Ca. Scalindua* (predominantly adapted to marine water environments) were enriched and grown as planktonic cells in membrane bioreactors (Supplementary Fig. 1a)¹⁵ Fluorescence in situ hybridization (FISH) showed that the anammox bacteria constituted >95% of the bioreactor's community (Supplementary Fig. 1b-g). Also, a previously enriched *K. stuttgartiensis* (predominantly adapted to freshwater environments) culture was used⁴."

Comment # 9: L 92-93. Basing reaction occurrence on visual observation is weak. Can GO conversion be chemically determined?

Response: We agree with the reviewer that visual inspection is not an appropriate method to suggest if a chemical reaction occurred or not. As explained in the manuscript and Fig. 1b, we also determined the conversion of graphene oxide (GO) to reduced graphene oxide (rGO) with Raman spectroscopy, which is one of the main techniques for characterizing graphene-based materials (Eigler et al., 2013; Wu et al., 2018):

Line 99: “Reduction of GO to rGO by anammox bacteria was further confirmed by Raman spectroscopy, where the formation of the characteristic 2D and D+D’ peaks of rGO¹⁷ were detected in the vials with anammox cells (Fig. 1b), whereas no peaks were detected in the abiotic control.”

Comment # 10: L195-196. The non-detection of known electrogenic bacteria cannot rule out the contribution of hitherto unknown electrogenic bacteria.

Response: We agree with the reviewer that the non-detection of known electrogenic bacteria cannot rule out the contribution of other unknown electroactive bacteria. However, as explained in our response to Comment # 2 above, anammox bacteria were solely responsible for NH₄⁺ oxidation and current generation and other (known or unknown) electroactive organisms were not involved in the process.

Comment # 11: The discussion on the reactions inside and outside the anammoxosome are lacking in detail. There is an abrupt mention of it acting as a barrier for electron transfer, without any perspective or further discussion.

Response: We thank the reviewer for pointing the importance of discussing the electron flow inside and outside the anammoxosome. Due to limitation of manuscript length, we only discussed the key steps in ammonium oxidation and EET in the main manuscript. However, we extensively discussed every step of the electron flow from ammonium oxidation in the anammoxosome, transfer through the membranes and extracellular electron transfer in the original supplementary discussion associated to the main manuscript. Please see sections “Putative EET-dependent anammox pathway”, “Respiratory complexes of anammox bacteria in EET-dependent anammox process” and “Central carbon metabolism of anammox bacteria in EET-dependent anammox process” in the supplementary discussion. We also summarized the electron flow and reactions of the process in supplementary Fig. 10, which is also referenced in the main manuscript.

Reviewer #3

Note to Reviewer # 3: Our responses to the comments have been divided and numbered to facilitate review. Reviewers' comments appear in *italics*, whereas our responses are non-italicized. All changes in the revised manuscript appear in red.

Comment # 1: Shaw et al. experimentally show that both multiple distant phylogenetic lineages of anammox bacteria (both marine and freshwater taxa) can couple NH₄⁺ oxidation and extracellular electron transfer (EET). This is demonstrated through a number of different experiments that included isotope chemistry, visualization, and sequencing. This work represents a significant advance in the field, working with bacterial organisms that are difficult to culture in highly controlled conditions. The authors also go attempt to better understand pathways that are used by these organisms to overcome the additional barrier of the anammoxosome compartment. While some aspects of the study such as the transcriptomics were not as successful as was likely hoped for, the study nonetheless represents elegant work and an advance in the field.

Response: We thank the reviewer for noting that the work represents a significant advance in the field. Also, we thank the reviewer for carefully going through our manuscript and providing constructive feedback to improve the quality of the manuscript. The following are our responses to the comments.

Comment # 2: Overall I think that it would be valuable if the author's provided some speculation on how likely it is for this process to occur in natural environments and implications of the occurrence of this reaction in natural environments. The engineering implications are much more well developed. Other minor comments are below.

Response: We thank the reviewer for this valuable suggestion. We now added further discussion in the revised manuscript on how likely EET-dependent anaerobic ammonium oxidation process may occur in natural environments and its implications:

Line 298: "Electrochemically active bacteria are typically found in environments devoid of oxygen or other soluble electron acceptors²⁷. Our results show a novel process of anaerobic ammonium oxidation coupled to EET-based respiration of carbon-based insoluble extracellular electron acceptor by both freshwater and marine anammox bacteria and suggest that this process may also occur in natural anoxic environments where soluble electron acceptors are not available. In environments such as anoxic sediments, microbial metabolism is limited by the diffusive supply of electron acceptors²⁸. Nitrogen loss by anammox and denitrification are expected to be limited by the diffusive flux of NO₂⁻ or NO₃⁻, and/or by O₂ diffusion that can be used by aerobic NH₄⁺ oxidizers. However, ³⁰N₂ production has been observed in ¹⁵NH₄⁺ incubations of sediments at depths below the penetration depth of NO₂⁻, NO₃⁻ and O₂ in marine^{11,28-30} and freshwater³¹ environments. These observations cannot be explained by conventional denitrification or anammox process. Interestingly, this phenomenon was observed in sediments rich in metal oxides or natural organic matter such as humic substances^{11,31}. EET-dependent anaerobic ammonium oxidation may play a role in such environments devoid of soluble electron acceptors such as NO₂⁻, NO₃⁻ and O₂. In natural environments, the NH₄⁺ used by anammox bacteria is derived from heterotrophic pathways of degradation of organic matter³², and therefore anammox cannot operate independently from the mineralization of organic matter³³. Ubiquitous high-molecular-weight organic compounds such as humic substances are known to act as terminal electron acceptors in anaerobic microbial respiration^{10,34}. Also, humic substances may serve as redox mediators between electron donors and poorly soluble metal oxides minerals in soils and sediments mediating dissimilatory metal oxide reduction³⁴. Therefore, humic substances present in natural organic matter may be a good candidate in natural environments of carbon-based electron acceptor for EET-based anaerobic

ammonium oxidation, or as redox mediator between anammox and iron oxides. These results offer a new perspective of a key player involved in the biogeochemical nitrogen cycle, which previously was believed to rely strictly on soluble electron acceptors for NH_4^+ oxidation. The fact that anammox bacteria can perform NH_4^+ oxidation coupled with EET, suggest that this process may have implications in the global nitrogen cycle by contributing to the nitrogen loss in environments where soluble electron acceptors are unavailable. Therefore, a better understanding of EET processes contributes to our understanding of the cycles that occur on our planet²⁷.”

Abstract comments

Comment # 3: Line 26: Who does “their genomes” refer to? Unclear from the sentence structure.

Response: We thank the reviewer for this observation. We now modified the text to improve clarity:

Line 25: **Anammox** genomes contain homologs of *Geobacter* and *Shewanella* cytochromes involved in extracellular electron transfer (EET).

Comment # 4: Line 42. What perception is challenged? Make this more clear.

Response: As suggested by the reviewer, we now modified the text to make it clearer:

Line 40: “To our knowledge, our results provide the first experimental evidence that marine and freshwater anammox bacteria can couple NH_4^+ oxidation with EET, which is a significant finding and challenges our perception of a key player of anaerobic oxidation of NH_4^+ in **biogeochemical nitrogen cycle, which previously was believed to rely strictly on soluble electron acceptors for NH_4^+ oxidation.**”

Comment # 5: Please provide a more concrete link to bigger picture implications. Why does this coupling matter?

Response: As suggested by the reviewer, we now added to the abstract a more concrete link to the implications of coupling anaerobic ammonium oxidation and EET:

Line 40: “To our knowledge, our results provide the first experimental evidence that marine and freshwater anammox bacteria can couple NH_4^+ oxidation with EET, which is a significant finding and challenges our perception of a key player of anaerobic oxidation of NH_4^+ in **biogeochemical nitrogen cycle, which previously was believed to rely strictly on soluble electron acceptors for NH_4^+ oxidation.** Also, with EET-dependent anammox it is possible to achieve complete NH_4^+ oxidation to N_2 at low applied voltage (0.3-0.6 V vs. standard hydrogen electrode; SHE) and without accumulation of NO_2^- and NO_3^- . These findings are promising in the context of implementing EET-dependent anammox process for energy efficient treatment of nitrogen using bioelectrochemical systems.”

Main text comments

Comment # 6: Line 48. Possibly change showed to suggested

Response: As suggested by the reviewer we modified the text as follows:

Line 54: “More than a decade ago, preliminary experiments **suggested** that...”

Comment # 7: Line 50-51. Possibly could shorten this sentence. However, the mechanism of this coupling reaction has remained unexplored to date.

Response: We modified the text to make it clearer to the reader. We now write:

Line 56: “However, **extracellular electron transfer (EET) activity and molecular mechanism of this coupling reaction** has remained unexplored to date.”

Comment # 8: Line 50. Why would the presence of a coupling of NH₄⁺ oxidation and extracellular electron transfer (EET) be important? It would be helpful here to expand on the potential broader significance of this reaction.

Response: We thank the reviewer for this valuable suggestion. Please note that the importance of this coupling and the broader significance of this reaction has been addressed in our response to Comment # 2 and Comment # 5 above.

Comment # 9: Lines 52-63. This section is confusing. It would be helpful to rework to shorten the text and clarify the main points.

Response: the background information in Lines 52-63 is important to highlight the novelty and motivation behind our work. We modified the paragraph to make it clearer to the reader. We now write:

Line 56: “However, **extracellular electron transfer (EET) activity and molecular mechanism of this coupling reaction** has remained unexplored to date. Further, these **tests with *K. stuttgartiensis* and *Scalindua*** could not discriminate between Fe(III) oxide reduction for nutritional acquisition (i.e., via siderophores) versus respiration through EET⁸. Therefore, these preliminary experiments **are not conclusive to determine** if anammox bacteria have EET capability or not.”

Comment # 10: Line 303. I would change obvious to another word, such as important

Response: As suggested by the reviewer we modified the text as follows:

Line 332: “These findings have **important** implications in energy-efficient treatment of...”

Comment # 11: Line 294. How likely is it that this reaction occurs/is prevalent in natural environments? What is an example of a natural anoxic environment where this process might occur and be important?

Response: We now added further discussion in the revised manuscript on how likely EET-dependent anaerobic ammonium oxidation process may occur in natural environments and its implication, and we gave examples of natural anoxic environments where this process may occur. Please note that this comment has been addressed in our response to Comment # 2 above.

Grammatical checks

Comment # 12: Line 45. Has to have

Response: As suggested by the reviewer we modified the text as follows:

Line 51: “Also, anammox bacteria **have** been extensively investigated...”

Comment # 13: Line 63. Is to has been

Response: As suggested by the reviewer we modified the text as follows:

Line 68: “It **has been** known for more than two decades...”

Comment # 14: Line 102. Are to were

Response: As suggested by the reviewer we modified the text as follows:

Line 107: "To determine if anammox bacteria **were** still dominant..."

Comment # 15: Line 117. Use

Response: We really appreciate it if the reviewer can clarify the grammatical mistake that he/she wants us to correct. The original text reads as follows "Therefore, we tested if anammox bacteria interact with electrodes via EET and use them as the sole electron acceptor in MEC."

Comment # 16: Also, multiple places are missing articles "a", "an", "the". Check the text throughout. e.g. Line 39. Add in "an electrode"

Response: As suggested by the reviewer we added the missing articles throughout the manuscript.

Comment # 17: Line 82. "an" insoluble

Response: As suggested by the reviewer we modified the text as follows:

Line 87: "...with the reduction of **an** insoluble extracellular electron acceptor."

References:

- C. Koch, F. Harnisch, Is there a Specific Ecological Niche for Electroactive Microorganisms? *ChemElectroChem*. **3**, 1282–1295 (2016).
- Hu Z, van Alen T, Jetten MSM, Kartal B. Lysozyme and penicillin inhibit the growth of anaerobic ammonium-oxidizing planctomycetes. *Appl Environ Microbiol* 2013; **79**: 7763–9
- J. Bin Wu, M. L. Lin, X. Cong, H. N. Liu, P. H. Tan, Raman spectroscopy of graphene-based materials and its applications in related devices. *Chem. Soc. Rev.* **47**, 1822–1873 (2018).
- Oshiki, M. *et al.* Nitrate-Dependent Ferrous Iron Oxidation by Anaerobic Ammonium Oxidation (Anammox) Bacteria. *Appl Environ Microbiol* 2013; **79**: 4087–4093.
- S. Eigler, S. Grimm, M. Enzelberger-Heim, P. Müller, A. Hirsch, Graphene oxide: Efficiency of reducing agents. *Chem. Commun.* **49**, 7391–7393 (2013).

REVIEWERS' COMMENTS:

Reviewer #1 (Remarks to the Author):

The authors addressed the comments fully sufficiently.

Reviewer #2 (Remarks to the Author):

Review 2 of 'Extracellular electron transfer-dependent anaerobic oxidation of ammonium by anammox bacteria'

Major comments

It is extremely disappointing to see that the authors have not made an earnest attempt to respond to the critical questions raised during the first round of reviews. Rather, the approach has been to simply repeat their initial assertions without a balanced scientific consideration of alternate possibilities suggested in the review. It is also utterly inappropriate to play the comments of one reviewer against that of another reviewer as the authors have done. The whole point of multiple reviews is to invite a broad set of views on any submitted work. It is suggested that the authors provide independent responses to comments by each reviewer.

Specific comments are presented below.

Response to comment 2. The authors first respond by saying that they did not use metatranscriptomics measurements to infer function but to compare the metabolic pathway of NH_4^+ oxidation. However, they are doing just that. Without getting buried in semantics, how is it possible to compare the metabolic pathway of NH_4^+ oxidation even possible (using mRNA measurements) without the basic inference of NH_4^+ oxidation function through the related transcripts? Perhaps the authors could clarify this directly without resorting to obfuscating terminology. Further below in the same response, the authors state that they used bioelectrochemistry and stable isotope analyses to reach their conclusion on the role of anammox bacteria. So, which one is it? Did the authors use mRNA, bioelectrochemistry and stable isotope analyses in combination? If so, why is it a problem to acknowledge a basic premise in the central dogma of molecular biology on the inability (consistently) of mRNA measurements to provide conclusive information on functional activity?

Response to comment 2. The comment raised the possibility that penicillin G may not have arrested the activity of all heterotrophic denitrifying organisms. While the authors acknowledge this in the response, there is no attempt to include this point as a qualifier in the manuscript. Again, this leads to an incomplete presentation of scientific possibilities and a biased cherry-picked discussion that conveniently presents a partial picture. This is unacceptable.

Response to comment 3. To clarify, this comment related to the production of nitrate by anammox bacteria and its use by nitrate reducing organisms. This is still not addressed.

Response to multiple comments. The authors keep repeating that they have provided 'solid evidence' but they have repeatedly failed to do so.

Reviewer #3 (Remarks to the Author):

We thank the authors for satisfactorily addressing comments/concerns raised in the first round of review.

Point-by-point response to Reviewer # 2

Note to Reviewer # 2: Our responses to the comments have been divided and numbered to facilitate review. Reviewers' comments appear in *italics*, whereas our responses are non-italicized. All changes in the revised manuscript appear in **red**.

Comment # 1: It is extremely disappointing to see that the authors have not made an earnest attempt to respond to the critical questions raised during the first round of reviews. Rather, the approach has been to simply repeat their initial assertions without a balanced scientific consideration of alternate possibilities suggested in the review.

Response: We thank Reviewer # 2 for reviewing our responses to the first round of revision. Also, we appreciate reviewer's valuable suggestions to improve our manuscript by including the importance of alternate possibilities suggested in the review. We have made all efforts to address the specific concerns (comments # 2 to #4) raised by the reviewer.

Comment # 2: Response to comment 2. The authors first respond by saying that they did not use metatranscriptomics measurements to infer function but to compare the metabolic pathway of NH₄⁺ oxidation. However, they are doing just that. Without getting buried in semantics, how is it possible to compare the metabolic pathway of NH₄⁺ oxidation even possible (using mRNA measurements) without the basic inference of NH₄⁺ oxidation function through the related transcripts? Perhaps the authors could clarify this directly without resorting to obfuscating terminology. Further below in the same response, the authors state that they used bioelectrochemistry and stable isotope analyses to reach their conclusion on the role of anammox bacteria. So, which one is it? Did the authors use mRNA, bioelectrochemistry and stable isotope analyses in combination? If so, why is it a problem to acknowledge a basic premise in the central dogma of molecular biology on the inability (consistently) of mRNA measurements to provide conclusive information on functional activity?

Response: We thank the reviewer for this valuable comment. In this study, every set of experiments (bioelectrochemistry, stable isotope and mRNA) provided a different information about anammox bacteria. For example, through bioelectrochemical analyses we reached to the conclusion that anammox bacteria is electrochemically active and are solely responsible for NH₄⁺ oxidation and current generation, and that other autotrophic or heterotrophic electroactive bacteria (known or unknown) were not involved in current generation, as described in the experimental evidences provided in the section "Electroactivity of anammox bacteria" of the manuscript. After confirming that anammox bacteria is electrochemically active, we conducted stable isotope experiments to better understand how NH₄⁺ is converted to N₂ by anammox bacteria in EET-dependent anammox process. Based on stable-isotope experiments we reached to the conclusion that NH₂OH, and not NO (intermediate in conventional anammox process), is an intermediate of EET-dependent anammox process, as described in the experimental evidences provided in the section "Molecular mechanism of EET-dependent anammox process" of the manuscript. Finally, we conducted genome-centric comparative transcriptomics analysis which revealed an alternative pathway for NH₄⁺ oxidation coupled to EET when an electrode is used as electron acceptor compared to NO₂⁻ as the electron acceptor.

To address reviewer's questions "*the authors state that they used bioelectrochemistry and stable isotope analyses to reach their conclusion on the role of anammox bacteria. So, which one is it? Did the authors use mRNA, bioelectrochemistry and stable isotope analyses in combination?*", we now

revised the manuscript to clarify to the readers what information was obtained from each set of experiments:

Line 252: “After confirming through bioelectrochemical analyses that anammox bacteria are electrochemically active, isotope labelling experiments were carried out to better understand how NH_4^+ is converted to N_2 by anammox bacteria in EET-dependent anammox process.”

Line 297: “In order to identify the possible pathways involved in NH_4^+ oxidation and electron flow through compartments (anammoxosome) and membranes (cytoplasm and periplasm) in EET-dependent anammox process (electrode poised at 0.6 V vs. SHE as electron acceptor) versus typical anammox process (i.e., nitrite used as electron acceptor), we conducted a genome-centric comparative transcriptomics analysis...”

Also, as suggested by the reviewer we now acknowledge the limitations of metatranscriptomics analysis for its inability to provide conclusive information on functional activity:

Line 297: “In order to identify the possible pathways involved in NH_4^+ oxidation and electron flow through compartments (anammoxosome) and membranes (cytoplasm and periplasm) in EET-dependent anammox process (electrode poised at 0.6 V vs. SHE as electron acceptor) versus typical anammox process (i.e., nitrite used as electron acceptor), we conducted a genome-centric comparative transcriptomics analysis (Supplementary discussion). Even though comparative transcriptomics analysis is a useful approach for exploring and detecting genes not previously known to play a role in adaptative responses to environmental changes, the levels of mRNA are not directly proportional to the expression level of the protein they encode, and therefore it is difficult to predict protein function and activity from quantitative transcriptome data. Accordingly, the results presented below from the comparative transcriptomics analysis should be read as hypothetical.”

Comment # 3: Response to comment 2. The comment raised the possibility that penicillin G may not have arrested the activity of all heterotrophic denitrifying organisms. While the authors acknowledge this in the response, there is no attempt to include this point as a qualifier in the manuscript. Again, this leads to an incomplete presentation of scientific possibilities and a biased cherry-picked discussion that conveniently presents a partial picture. This is unacceptable.

Response: We now included our response on Penicillin G in the revised manuscript.

Line 221: “In addition, NH_4^+ oxidation and current production were not affected by the addition of Penicillin G (Supplementary Fig. 4), a compound that has inhibitory effects in some heterotrophs, but it does not have any observable short-term effects on anammox activity^{23,24}. Similar results were obtained with *Ca. Scalindua* and *K. stuttgartiensis* (data not shown). As pointed above, one of the limitations of Penicillin G is that it does not arrest the activity of all heterotrophs. Despite this limitation, the role of heterotrophs in current production was excluded in another control experiment. Since no exogenous organic carbon was added to the MEC reactors, the only source of organics for heterotrophic organisms was through endogenous decay. However, there was no current generation in the absence of NH_4^+ (Fig. 2d), suggesting the lack of involvement of heterotrophic electroactive bacteria in the process.”

Comment # 4: Response to comment 3. To clarify, this comment related to the production of nitrate by anammox bacteria and its use by nitrate reducing organisms. This is still not addressed.

Response: We thank the reviewer for clarifying this comment. We would like to emphasize that no nitrate was produced in the EET-dependent anammox process. This was further clarified in the revised manuscript, with the following text, which appears in red.

Line 164: “When the exogenous electron acceptor (i.e., NO_2^-) was completely removed from the feed, anammox cells began to form a biofilm on the surface of the electrodes (Supplementary Fig. 1i) and current generation coupled to NH_4^+ oxidation was observed in the absence of NO_2^- (Fig. 2a). Further, NO_2^- and NO_3^- were below the detection limit at all time points when the working electrode was used as the sole electron acceptor, suggesting that NO_2^- and NO_3^- did not play an apparent role in the process.”

Line 371: “Also, compared to conventional anammox process, EET-dependent anammox process achieved complete removal of NH_4^+ (at low and high concentrations) to nitrogen gas with no accumulation of NO_2^- or NO_3^- or the production of the greenhouse gas N_2O . In conventional anammox process (i.e., when NO_2^- is used as the electron acceptor), NO_3^- is generated as a result of the oxidation of NO_2^- by anammox bacteria. Consequently, the effluent from conventional anammox process for wastewater treatment requires further polishing by nitrate reducing organisms before discharge. In contrast, as shown by our results, since NO_2^- was not added in the EET-dependent anammox process, no production of NO_3^- was detected.